# Bronchus-associated macrophages efficiently capture and present soluble inhaled antigens and are capable of local Th2 cell activation

Xin-Zi Tang[1,2,3†], Lieselotte S M Kreuk[1,2‡], Cynthia Cho[1,2§], Ross J Metzger[4#], Christopher D C Allen[1,2,4*]

[1]Cardiovascular Research Institute, University of California, San Francisco, San Francisco, United States; [2]Sandler Asthma Basic Research Center, University of California, San Francisco, San Francisco, United States; [3]Biomedical Sciences Graduate Program, University of California, San Francisco, San Francisco, United States; [4]Department of Anatomy, University of California, San Francisco, San Francisco, United States

**\*For correspondence:**
Chris.Allen@ucsf.edu

**Present address:** †Molecular Engineering Laboratory, Institute of Molecular and Cell Biology, Agency for Science, Technology and Research, Biopolis, Singapore; ‡Department of Microbiology, University of Washington School of Medicine, Seattle, United States; §Nkarta Therapeutics, South San Francisco, San Francisco, United States; #Department of Pediatrics (Cardiology), Stanford University School of Medicine, Stanford, United States

**Competing interest:** The authors declare that no competing interests exist.

**Abstract** In allergic asthma, allergen inhalation leads to local Th2 cell activation and peribronchial inflammation. However, the mechanisms for local antigen capture and presentation remain unclear. By two-photon microscopy of the mouse lung, we established that soluble antigens in the bronchial airway lumen were efficiently captured and presented by a population of CD11c⁺ interstitial macrophages with high CX3CR1-GFP and MHC class II expression. We refer to these cells as Bronchus-Associated Macrophages (BAMs) based on their localization underneath the bronchial epithelium. BAMs were enriched in collagen-rich regions near some airway branchpoints, where inhaled antigens are likely to deposit. BAMs engaged in extended interactions with effector Th2 cells and promoted Th2 cytokine production. BAMs were also often in contact with dendritic cells (DCs). After exposure to inflammatory stimuli, DCs migrated to draining lymph nodes, whereas BAMs remained lung resident. We propose that BAMs act as local antigen presenting cells in the lung and also transfer antigen to DCs.

## Editor's evaluation

The study here presented interesting evidence on the nature of a cell population responsible for antigen uptake and presentation in the airway and how this cell population could stimulate a Th2 response locally. Authors performed multi photon imaging of live tissue or 3D reconstruction of thick lung sections from various transgenic mouse models. The manuscript provides an elegant set of data that consolidate the study focusing on BAMs, their strategic positioning near the airways, the characterization of antigen capture and presentation, as well as migratory properties, as compared to DCs.

## Introduction

Immune surveillance of barrier surfaces is essential for the maintenance of tissue homeostasis and defense against pathogens. Within the lung, the conducting bronchial airways represent one of the first sites of exposure to inhaled substances and infectious agents. In allergic asthma, the inhalation of harmless environmental allergens leads to peribronchial inflammation, in which the local activation

of type 2 helper T cells (Th2 cells) results in effector cytokine production that promotes pathogenesis (*Holgate, 2012*; *Lambrecht et al., 2019*). The Th2 cells are activated by antigen presenting cells (APCs) that survey the epithelial barrier and capture and present antigen. However, the types of APCs contributing to surveillance of the conducting airways are not fully characterized, and the anatomical organization that couples antigen capture to Th2 cell activation remains unclear.

The lung tissue regions around the conducting airways may be ideal sites for antigen surveillance and the initiation of immune responses. Unlike alveolar spaces, the conducting airways do not participate in gaseous exchange but serve to conduct and filter air. As inspired air passes through airway branchpoints, changes in the direction of airflow promote the deposition of particulates and the dissolution of soluble substances in the overlying mucus layer (*Carvalho et al., 2011*; *Clarke and Yeates, 1994*). In allergic asthma, eosinophils and lymphocytes are recruited to regions around the conducting airways, resulting in peribronchial inflammation (*Jeffery, 1992*). In several mouse models of airway infections, T and B cells accumulate in organized foci near the airways, forming induced Bronchus-Associated Lymphoid Tissue (BALT) structures (*Fleige and Förster, 2017*; *Randall, 2010*). The accumulation of immune cells in tissue regions near the conducting airways may allow streamlined access to inhaled antigens that have become deposited.

The mechanism by which antigen capture and immune activation occurs at the airways is unclear. Most studies have focused on the potential contributions of dendritic cells (DCs) to allergic inflammation in the lung (*Wikstrom and Stumbles, 2007*), based initially on observations that MHC-II⁺ cells with dendritic morphology were abundant along both human and rodent airways at baseline (*Demedts et al., 2005*; *Huh et al., 2003*; *Lambrecht et al., 1998*; *von Garnier et al., 2007*). Flow cytometric analysis of lung-draining lymph nodes, dissociated lung tissue enriched for the conducting airways, and the trachea also showed that DCs, gated as CD11c⁺ MHC-II⁺ cells, were able to capture, process, and present fluorescent ovalbumin (OVA) administered via the airways (*del Rio et al., 2007*; *Fear et al., 2011*; *Hammad et al., 2010*; *Huh et al., 2003*; *von Garnier et al., 2007*). The uptake of OVA by CD11c⁺ MHC-II⁺ cells throughout the lung has also been observed by confocal microscopy (*Hoffmann et al., 2018*) however, two-photon microscopy studies of CD11c-YFP reporter mice found that DCs near the conducting airways did not detectably capture soluble antigens (OVA or dextran), or particulate antigens in the form of beads (*Thornton et al., 2012*; *Veres et al., 2011*). CD11c-YFP⁺ DCs were reported to capture beads at the alveoli in a mouse model of asthma, and then were proposed to migrate to tissue regions proximal to the bronchial airways, where T cells are activated (*Thornton et al., 2012*). This mechanism might be relevant for the uptake and presentation of particulate antigens that reach the alveoli, but whether soluble antigen can be directly captured and presented by APCs at the conducting airways remains unclear. The presentation of soluble antigens is pertinent because it was reported that individuals allergic to aeroallergens had an increased frequency of Th2 cells that preferentially recognized soluble antigen components of aeroallergens, whereas regulatory T cells preferentially recognized particulate antigen components (*Bacher et al., 2016*). Developing a better understanding of the uptake and presentation of soluble antigens may therefore be of significant relevance to Th2 cell responses in asthma.

In the intestine, soluble antigen in the lumen is efficiently captured by CX3CR1⁺ macrophages in the lamina propria (*Farache et al., 2013a*; *Farache et al., 2013b*; *Mazzini et al., 2014*; *Schulz et al., 2009*). These intestinal macrophages were found to be distinct from conventional DCs (cDCs) based on functional and behavioral differences and were reported to contribute to local intestinal immunity (*Hadis et al., 2011*; *Mazzini et al., 2014*; *Schreiber et al., 2013*; *Schulz et al., 2009*). Recent gene expression and microscopy studies of the lung have identified CX3CR1⁺ interstitial macrophages (IMs) that are distinct from alveolar macrophages (AMs). Notably, subpopulation(s) of these IMs express many of the surface markers previously thought to be exclusive to DCs, such as MHC-II and CD11c (*Chakarov et al., 2019*; *Gautier et al., 2012*; *Gibbings et al., 2017*; *Sabatel et al., 2017*; *Schlitzer et al., 2013*; *Schyns et al., 2019*; *Ural et al., 2020*). While most studies on resident tissue macrophages have focused on comparing various macrophage populations to each other, how lung DCs and IMs compare in their behaviors and functions is yet unclear.

Here, we show that in the steady state in naïve mice, lung myeloid APCs are primarily positioned underneath the bronchial airway epithelium and are strategically enriched near some airway branchpoints where inhaled allergens are likely to deposit. We found that at these regions, the majority of inhaled soluble antigens were captured by CX3CR1⁺ CD11c⁺ MHC-II⁺ IMs rather than DCs. We

observed frequent contacts between CX3CR1+ CD11c+ MHC-II+ IMs and DCs, suggesting the possibility of antigen transfer to DCs. Yet CX3CR1+ CD11c+ MHC-II+ IMs themselves were capable of antigen processing and presentation, activated T cells in vitro, and interacted with T cells recruited to the airways in an asthma model. Unlike DCs, CX3CR1+ CD11c+ MHC-II+ IMs did not migrate to the draining lymph node and remained lung resident. We therefore propose that CX3CR1+ CD11c+ MHC-II+ IMs located near the bronchial airways are tissue-resident sentinel APCs that are able to drive local allergic inflammation, whereas DCs may be more critical for priming T cell responses in lymph nodes.

## Results

### Myeloid cells are enriched near bifurcating airways prior to the induction of inflammation

Peribronchial inflammation is a hallmark of asthma that is recapitulated in mouse models of allergic airway disease. Foci of leukocytes can be observed near the bronchi that include numerous eosinophils. In our own studies of mouse models of allergic airway disease, such as in a classical model with OVA as an allergen (*Daubeuf and Frossard, 2013*), we observed in thin cryosections that these foci were not evenly distributed along the airway, but rather were typically located near airway branchpoints and associated with large blood vessels (*Figure 1A*). These areas were further visualized in 3D in agarose-inflated, thick lung sections that were precision cut with a vibratome and imaged by two-photon laser scanning microscopy (hereafter referred to as two-photon microscopy). Bronchial airway epithelial cells were revealed by GFP expression when we crossed Sonic hedgehog (Shh)-Cre mice (*Harfe et al., 2004*) to ROSA26-mTmG reporter mice, in which Cre-mediated recombination leads to the expression of membrane-bound GFP (*Muzumdar et al., 2007*). Alveolar epithelial cells also express GFP in these mice, with alveolar type II cells appearing more prominently due to their cuboidal shape, whereas all other cells express membrane-bound tdTomato. Collagen was detected by second harmonic generation. Our imaging revealed regions near some airway branchpoints with dense accumulations of collagen surrounding blood vessels (*Figure 1B* and *Video 1*). These regions likely represent bronchovascular bundles which are known to contain bronchi and arteries in a common adventitial sheath (*Townsley, 2012*). Bifurcating blood vessels, visualized by membrane-bound tdTomato, were also visible at some collagen-rich sites in which the airways appeared to undergo planar bifurcation (*Metzger et al., 2008*; *Figure 1B* and *Video 1*).

Various myeloid cells are known to accumulate in peribronchial regions during inflammation. To visualize myeloid cell infiltration, we imaged thick lung sections from mice expressing the MacGreen (CSF1R-GFP) reporter (*Sasmono et al., 2003*) by two-photon microscopy. In MacGreen mice, we had previously observed that most myeloid cells including macrophages, monocytes, neutrophils, and CD11b+ DCs brightly expressed the GFP reporter, although eosinophils exhibited much lower GFP expression (data not shown). As some B cells were also found to express GFP in these mice (unpublished observations), we had crossed the MacGreen mice to B-cell-deficient μMT mice so that GFP expression was restricted to myeloid cells. At the collagen-rich regions near some bifurcating airways, we observed dense accumulations of myeloid cells in both OVA and house dust mite (HDM) models of allergic airway disease (*Figure 1C and D*). Interestingly, in naïve mice, a sparse population of MacGreen+ cells was also observed at these collagen-rich regions near bifurcating airways (*Figure 1E*), suggesting that myeloid cells may localize to these areas prior to the formation of foci of leukocytes during inflammation. Notably, these myeloid cells were best visualized in naïve mice in thick lung sections by two-photon microscopy, because in thin cryosections the very low density of cells made it difficult to discern their distribution.

To further evaluate the types of myeloid cells located at these sites in the steady state, we imaged IMs and DCs using the CX3CR1-GFP and CD11c-YFP reporters, respectively, in thick lung sections by two-photon microscopy (*Gibbings et al., 2017*; *Jung et al., 2000*; *Lindquist et al., 2004*; *Rodero et al., 2015*; *Thornton et al., 2012*; *Veres et al., 2011*). In order to specifically visualize IMs and DCs, our analysis focused on cells with bright expression of these fluorescent reporters using limiting laser power, as other cell types have weaker expression of these fluorescent reporters. For example, AMs, which express high amounts of surface CD11c protein, are known to have low expression of the CD11c-YFP transgene in naïve mice (*Rodero et al., 2015*; *Thornton et al., 2012*). We simultaneously visualized airway epithelial cells with the E-cadherin-mCFP reporter (*Snippert et al., 2010*), or with

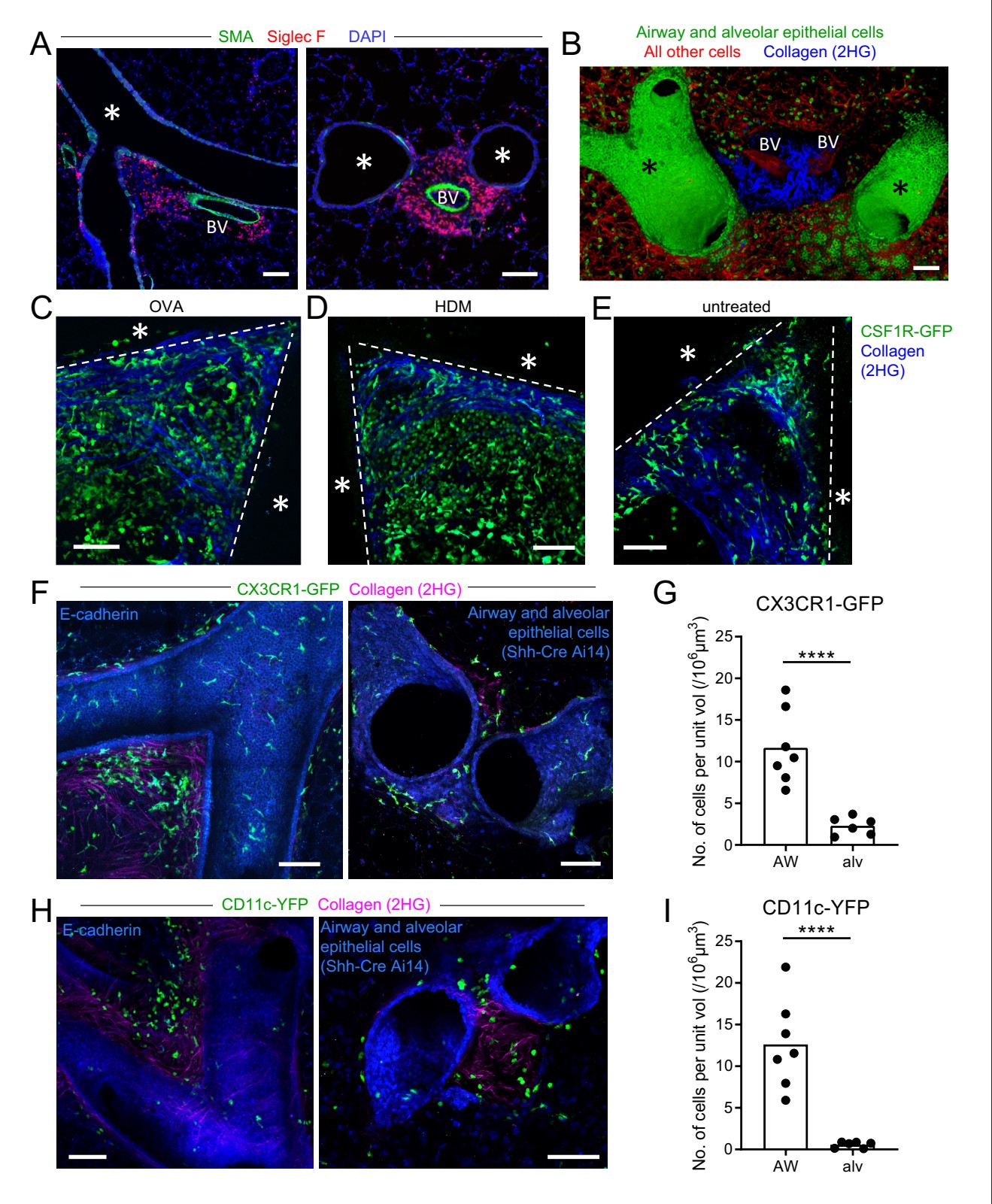

**Figure 1.** Myeloid cells are strategically positioned for surveillance of the airways. (**A**) Representative immunofluorescence staining of thin cryostat sections of the lung from wild-type mice sensitized and challenged with OVA in a classical OVA model of allergic airway disease. Sections were stained with an antibody to smooth muscle actin (SMA, green) to differentiate between airways and blood vessels, and with an antibody to Siglec F (red) to visualize tissue eosinophils (Siglec-F$^{bright}$) and AMs (Siglec-F$^{int}$). Sections were stained with DAPI (blue) to visualize all nuclei. Airways (*) and blood

*Figure 1 continued on next page*

*Figure 1 continued*

vessels (BV) are labeled. (**B**) Representative 3D fluorescence opacity rendering of branching airways (*) and blood vessels (BV). The image is from a 516 µm z-stack collected by two-photon microscopy of a precision-cut thick vibratome slice of the lung from a naïve Shh-Cre ROSA26-mTmG mouse. Bronchial airway and alveolar epithelial cells express membrane-bound GFP (green), whereas all other cells express tdTomato (red). Collagen (blue) was visualized by second harmonic generation (2HG). (**C, D, E**) Representative two-photon microscopy images of precision-cut thick vibratome slices of the lungs of MacGreen (CSF1R-GFP, green) µMT mice that were treated with OVA (**C**) or HDM (**D**) allergic airway disease models, or were untreated (**E**). Maximum Intensity Projections (MIPs) representing z-stacks of 165 µm (**C**), 122.5 µm (**D**), and 127.5 µm (**E**) are shown. The bronchial airways were visualized by transmitted illumination and the approximate boundaries are denoted by dashed lines, with the airway lumen denoted by *. Collagen (blue) was visualized by 2HG. (**F**) Representative two-photon microscopy images of thick lung slices from mice expressing CX3CR1-GFP (green) together with E-cadherin-mCFP (left, blue) or Shh-Cre Ai14 (right, blue) to label airway epithelial cells. Collagen (magenta) was visualized by 2HG. MIPs representing z-stacks of 210 µm (left) and 142 µm (right) are shown. (**G**) Quantification of the density of CX3CR1-GFP[bright] cells located in tissue regions near the bronchial airways (AW) or alveoli (alv). (**H**) Representative two-photon microscopy images of thick lung slices from mice expressing CD11c-YFP (green) together with E-cadherin-mCFP (left, blue) or Shh-Cre Ai14 (right, blue) to label airway epithelial cells. Collagen (magenta) was visualized by 2HG. MIPs representing z-stacks of 231 µm (left) and 285 µm (right) are shown. (**I**) Quantification of the density of CD11c-YFP[+] cells in tissue regions near the bronchial airways (AW) or alveoli (alv). Data are representative of 3 experiments (**A**), 4 experiments (**B**), 3 experiments (**C**), 2 experiments (**D**) >5 experiments (**E**), and ≥3 experiments (**F, H**). Images in (**B–H**) are stitched from tiled images. Each data point in (**G, I**) represents the quantification of cell density in one region of interest collected from six tiled images (AW) and five tiled images (alv) from five different lung lobes from four mice across three experiments. ****$p<0.0001$ (t-test). Scale bars, 100 µm.

a combination of Shh-Cre with a ROSA26-loxP-STOP-loxP-tdTomato (Ai14) reporter (*Madisen et al., 2010*), such that airway epithelial cells expressed mCFP or tdTomato, respectively. In naïve mice, we observed clusters of bright CX3CR1-GFP[+] IMs and CD11c-YFP[+] DCs at some airway bifurcations in collagen-rich regions (*Figure 1F and H*, *Videos 2 and 3*). Scattered bright CX3CR1-GFP[+] IMs were also observed underneath the entire bronchial airway epithelium (*Figure 1F* and *Video 2*). Bright CX3CR1-GFP[+] cells were more abundant at the airways than at the alveoli (*Figure 1G*), and bright CD11c-YFP[+] DCs were only rarely observed at the alveoli in naïve mice (*Figure 1I*). Taken together, these findings indicate that in the steady-state in the absence of inflammation, both IMs and DCs are located near bronchial airways and may be particularly enriched in collagen-rich regions near bifurcations. These regions correspond to the sites where eosinophils and other leukocytes accumulate in allergic airway disease, suggesting a potential role for these IMs and/or DCs in the initiation of the local inflammatory response.

## CX3CR1-GFP[+] macrophages capture antigen from the bronchial airways

Our above findings regarding the positioning of CX3CR1-GFP[+] IMs and CD11c-YFP[+] DCs at some airway branchpoints may be significant, as antigen is likely to deposit on the walls of the airways at branchpoints where airflow changes directions. In addition, the distribution of CX3CR1-GFP[+] IMs underneath the entire bronchial airway epithelium suggests that these cells may generally be involved in surveillance of the airways. We next considered whether inhaled soluble antigens could be directly captured by the CX3CR1-GFP[+] IMs and CD11c-YFP[+] DCs located near the bronchial airways. In order to visualize the uptake of soluble protein antigens from the airway lumen, we instilled fluorescent OVA intranasally and prepared precision-cut lung slices 1–2 hr later

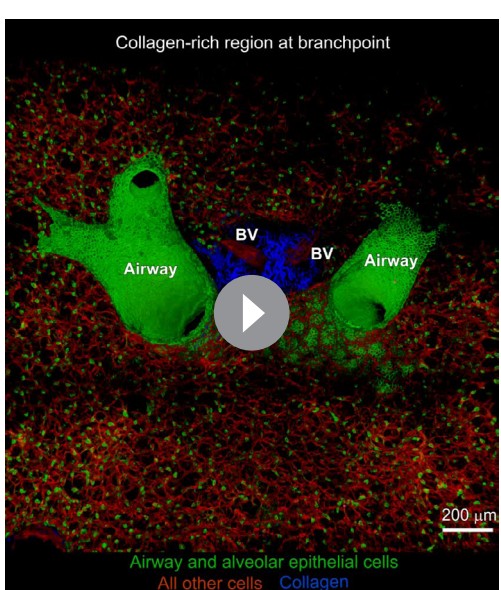

**Video 1.** Collagen-rich region at branchpoint. The video is a 3D rendering of the image depicted in Figure 1B, showing branching airways and blood vessels (BV) in the lung of a naïve Shh-Cre ROSA26-mTmG mouse. The image was stitched from tiled images collected by two-photon microscopy of a precision-cut thick vibratome section. Bronchial airway and alveolar epithelial cells express membrane-bound GFP (green), whereas all other cells express tdTomato (red). Collagen (blue) was visualized by second harmonic generation.

https://elifesciences.org/articles/63296/figures#video1

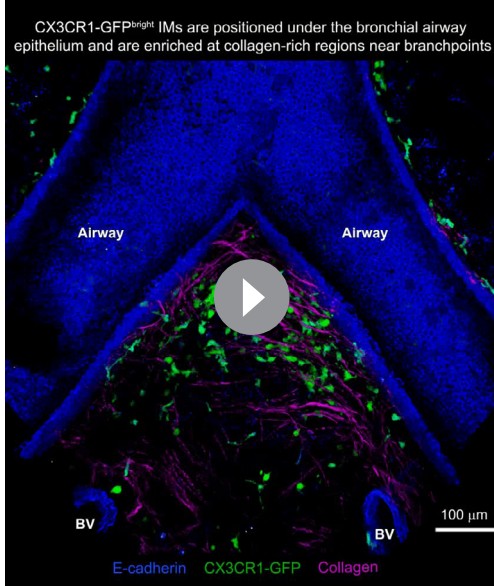

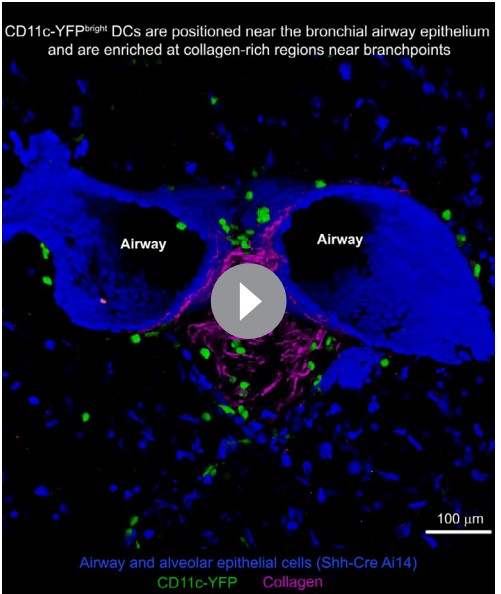

**Video 2.** CX3CR1-GFP[bright] IMs are positioned under the bronchial airway epithelium and are enriched at collagen-rich regions near branchpoints. The video is a 3D rendering of the image depicted in Figure 1F (left panel), showing branching airways and blood vessels (BV) in the lung of a naïve mouse expressing CX3CR1-GFP (green) to visualize IMs, together with E-cadherin-mCFP (blue) to visualize epithelial cells. Collagen (magenta) was visualized by second harmonic generation. The image was stitched from tiled images collected by two-photon microscopy of a precision-cut thick vibratome section. The video begins by showing the lumen of the airways with opacity rendering and then the image rotates to show CX3CR1-GFP[bright] IMs positioned underneath the airway epithelium. Finally, a zoomed in view of CX3CR1-GFP[bright] IMs in the collagen-rich region at the branchpoint is shown.
https://elifesciences.org/articles/63296/figures#video2

**Video 3.** CD11c-YFP[bright] DCs are positioned near the bronchial airway epithelium and are enriched at collagen-rich regions near branchpoints. The video is a 3D rendering of the image depicted in Figure 1H (right panel), showing branching airways in the lung of a naïve mouse expressing CD11c-YFP (green) to visualize DCs, together with Shh-Cre and the Ai14 (tdTomato) reporter to show airway and alveolar epithelial cells (blue). Collagen (magenta) was visualized by second harmonic generation. The image was stitched from tiled images collected by two-photon microscopy of a precision-cut thick vibratome section.
https://elifesciences.org/articles/63296/figures#video3

for imaging by two-photon microscopy. A short time point was chosen to ensure that the antigen capturing cells identified had not migrated from a different anatomical location, as had been proposed by *Thornton et al., 2012*. Airway epithelial cells were visualized by the E-cadherin-mCFP[+] reporter. Our analysis revealed numerous bright CX3CR1-GFP[+] IMs located near the bronchial airways that had taken up fluorescent OVA (*Figure 2A and B*). These bright OVA[+] CX3CR1-GFP[+] IMs had a dendritic morphology, extending long processes under the bronchial airway epithelium. As expected, soluble antigen was also taken up at the alveoli, primarily by AMs which had a distinct spherical appearance (*Figure 2—figure supplement 1*). Thus, CX3CR1-GFP[+] IMs with a dendritic morphology readily captured soluble antigen from the bronchial airway lumen and were distinct from AMs.

In contrast to the CX3CR1-GFP[+] IMs near the bronchial airways, the CD11c-YFP[+] DCs at these sites did not visibly appear to capture OVA by microscopy (*Figure 2C*). To confirm preferential uptake in CX3CR1-GFP[+] IMs rather than CD11c-YFP[+] DCs, we administered fluorescent OVA intranasally to mice simultaneously expressing both CX3CR1-GFP and CD11c-YFP reporters. This combination of fluorescent reporters has been helpful in the intestine to distinguish DC and macrophage subsets (*Farache et al., 2013a*; *McDole et al., 2012*), and we optimized methodology to minimize the spectral overlap of GFP and YFP by two-photon microscopy (see Methods section for details). Our imaging of dual reporter mice confirmed that OVA was preferentially captured by bright CX3CR1-GFP[+] IMs rather than bright CD11c-YFP[+] DCs (*Figure 2D*). Quantification of OVA uptake indicated that CD11c-YFP[+]

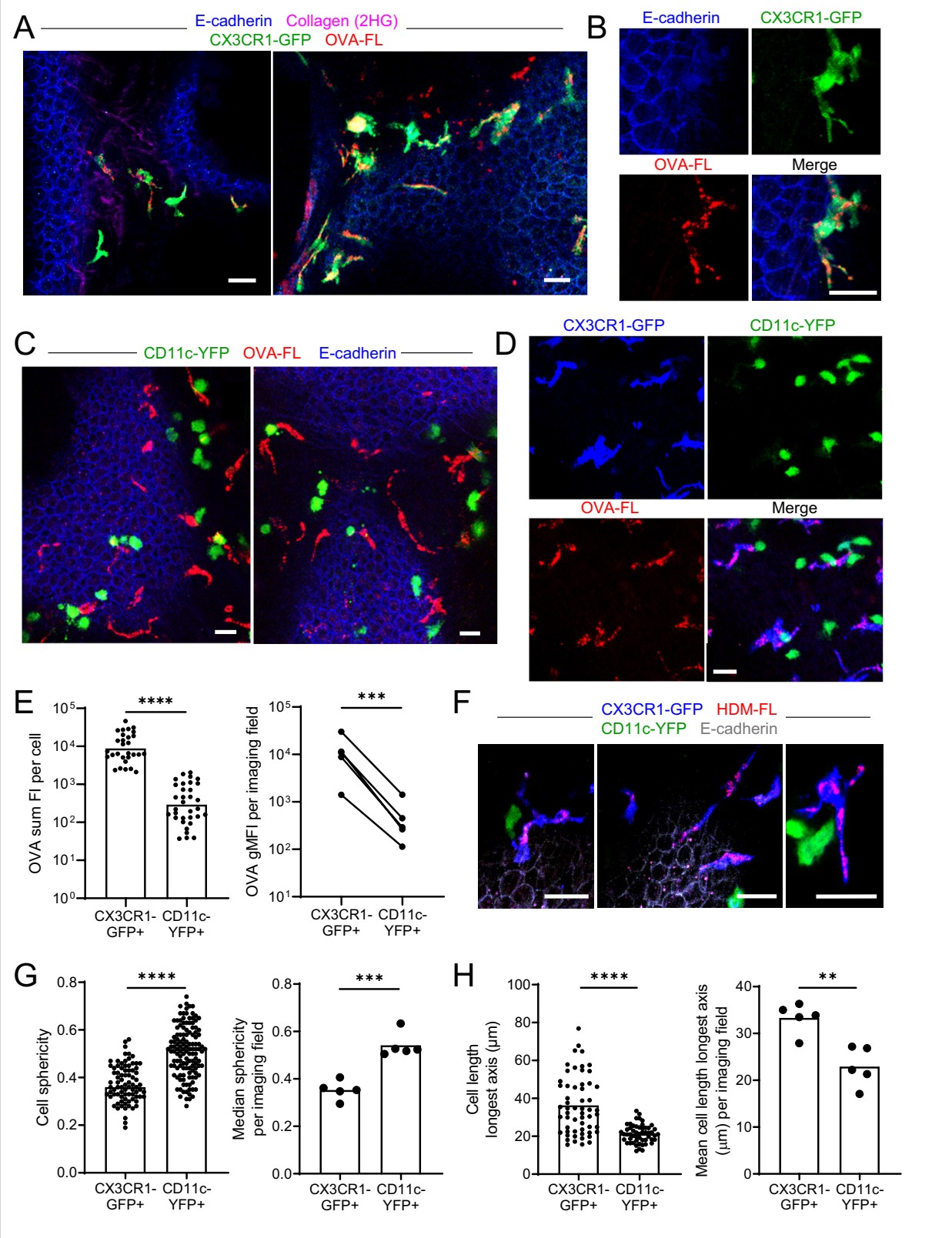

**Figure 2.** Antigen administered via the airways is captured by CX3CR1-GFP[bright] IMs. (**A–E**) Fluorescent OVA (OVA-FL, red) was administered by intranasal droplet to naïve mice 45 min-2 hr prior to lung isolation. Precision-cut, thick vibratome sections were then imaged by two-photon microscopy. Shown are representative images of regions near the bronchial airways from mice expressing CX3CR1-GFP (green) and E-cadherin-mCFP (blue) (**A, B**), CD11c-YFP (green) and E-cadherin-mCFP (blue) (**C**) and CX3CR1-GFP (blue) and CD11c-YFP (green) (**D**). In (**B**) and (**D**), single color and merged color

*Figure 2 continued on next page*

*Figure 2 continued*

images of the regions are shown. Collagen (magenta) was visualized in (**A**) by 2HG. MIPs representing z-stacks of 6 µm (**A**, left), 21 µm (**A**, right), 9 µm (**B**), 24 µm (**C**, left), 22 µm (**C**, right), and 24 µm (**D**) are shown. (**E**) Quantification of OVA-FL uptake by CX3CR1-GFP⁺ versus CD11c-YFP⁺ cells; shown are the sum of the fluorescence intensity (FI) per cell (left) for a representative imaging field, and geometric mean fluorescence intensity (gMFI) per imaging field (right). (**F**) Fluorescent HDM extract (HDM-FL, red) was instilled intranasally to naïve mice expressing CX3CR1-GFP (blue), CD11c-YFP (green), and E-cadherin-mCFP (gray). MIPs representing z-stacks of 12–13 µm are shown. (**G, H**) Analysis of the shapes of CX3CR1-GFP⁺ and CD11c-YFP⁺ cells in naïve mice by quantifying sphericity (**G**) and the length of the longest cell axis (**H**). Data are shown for individual cells as well as the median (**G**) or mean (**H**) of cells in each imaging field analyzed. Data are representative of five experiments (**A–B, D**); two experiments (**C,F**); or were analyzed from five experiments (**E**, right), or four experiments (**G–H**). ** p<0.01, *** p<0.001, **** p<0.0001 (t-test). Scale bars, 20 µm.

The online version of this article includes the following figure supplement(s) for figure 2:

**Figure supplement 1.** AMs in CX3CR1-GFP mice appear distinct from BAMs.

**Figure supplement 2.** Characterization of the relative intensity of CX3CR1-GFP versus CD11c-YFP fluorescence.

**Figure supplement 3.** OVA-capturing BAMs do not express the CD11c-YFP reporter.

---

DCs captured much smaller amounts of OVA than CX3CR1-GFP⁺ IMs (*Figure 2E*). We also observed preferential uptake of fluorescently labelled extracts from HDM, a relevant aeroallergen, by bright CX3CR1-GFP⁺ IMs (*Figure 2F*). Our imaging also confirmed that bright CX3CR1-GFP⁺ and bright CD11c-YFP⁺ cells were largely non-overlapping populations (*Figure 2D and F*). When we increased the image contrast, we observed that some of the bright CX3CR1-GFP⁺ cells did have dim expression of CD11c-YFP, and vice versa, as expected from previous studies (*Jung et al., 2000*; *Sen et al., 2016*; *Figure 2—figure supplement 2A*). Occasionally, we also observed rare cells that were bright for both the CX3CR1-GFP and CD11c-YFP reporters, yet the majority of cells in the naïve state appeared to be bright for only one of these reporters suggesting they were distinct cell types (*Figure 2—figure supplement 2B*). Indeed, CX3CR1-GFP⁺ IMs had a dendritic morphology as we had noted above, whereas CD11c-YFP⁺ DCs were rounder and did not appear to extend long processes. We quantified these morphologic features by measuring sphericity as well as the length of the longest axis of each cell to characterize the dendritic projections. CX3CR1-GFP⁺ IMs had lower sphericity values (*Figure 2G*) and increased mean lengths of their longest axes (*Figure 2H*) compared to CD11c-YFP⁺ DCs. The disparity in morphology and antigen uptake capacity suggests that CX3CR1-GFP⁺ IMs and CD11c-YFP⁺ DCs are two distinct cell populations present in airway-adjacent areas in the steady state, and that CX3CR1-GFP⁺ IMs are the primary antigen capturing cell type at the airways.

We confirmed these microscopy observations by flow cytometric analysis of CX3CR1-GFP versus CD11c-YFP reporter mice after instillation of fluorescent OVA. This analysis confirmed that the average intensity of fluorescent OVA was over three-fold brighter in CX3CR1-GFP^bright IMs compared with CD11c-YFP^bright DCs (*Figure 2—figure supplement 3A and B*). Microscopy detects a comparably narrower dynamic range of fluorescent signals than flow cytometry, and thus we most likely only visualized CX3CR1-GFP^bright IMs that carried the most OVA. Indeed, when we gated specifically on OVA^bright MHC-II⁺ cells by flow cytometry, the majority of cells were CX3CR1-GFP^bright but not CD11c-YFP^bright (*Figure 2—figure supplement 3C and D*). Therefore, our results show that DCs also take up soluble antigen from the bronchial airway lumen, but that the majority of antigen is taken up by a distinct population of CX3CR1-GFP⁺ IMs.

## CX3CR1-GFP^bright antigen capturing cells are Bronchus-Associated Macrophages (BAMs)

We next evaluated whether the CX3CR1-GFP⁺ cells that captured soluble inhaled antigen expressed MHC-II and confirmed their macrophage identity by flow cytometry. This was important given that CX3CR1-GFP is also expressed in other cell types, such as monocytes, a subset of DCs, and some lymphocytes (*Jung et al., 2000*). For this analysis, we excluded AMs using the marker Siglec-F, as these cells often confound the gating of GFP⁺ cells by flow cytometry due to their autofluorescence, yet based on their round morphology and alveolar localization were clearly distinct from the antigen-capturing CX3CR1-GFP⁺ cells near the airways (*Figure 2—figure supplement 1*). After administering fluorescent OVA intranasally and then isolating and enzymatically digesting the lungs, our flow cytometric analysis showed that the CX3CR1-GFP^bright OVA^bright cells were predominantly MHC-II^hi and CD11c^mid (*Figure 3A*). The presence of these cells in naïve mice and high MHC-II expression

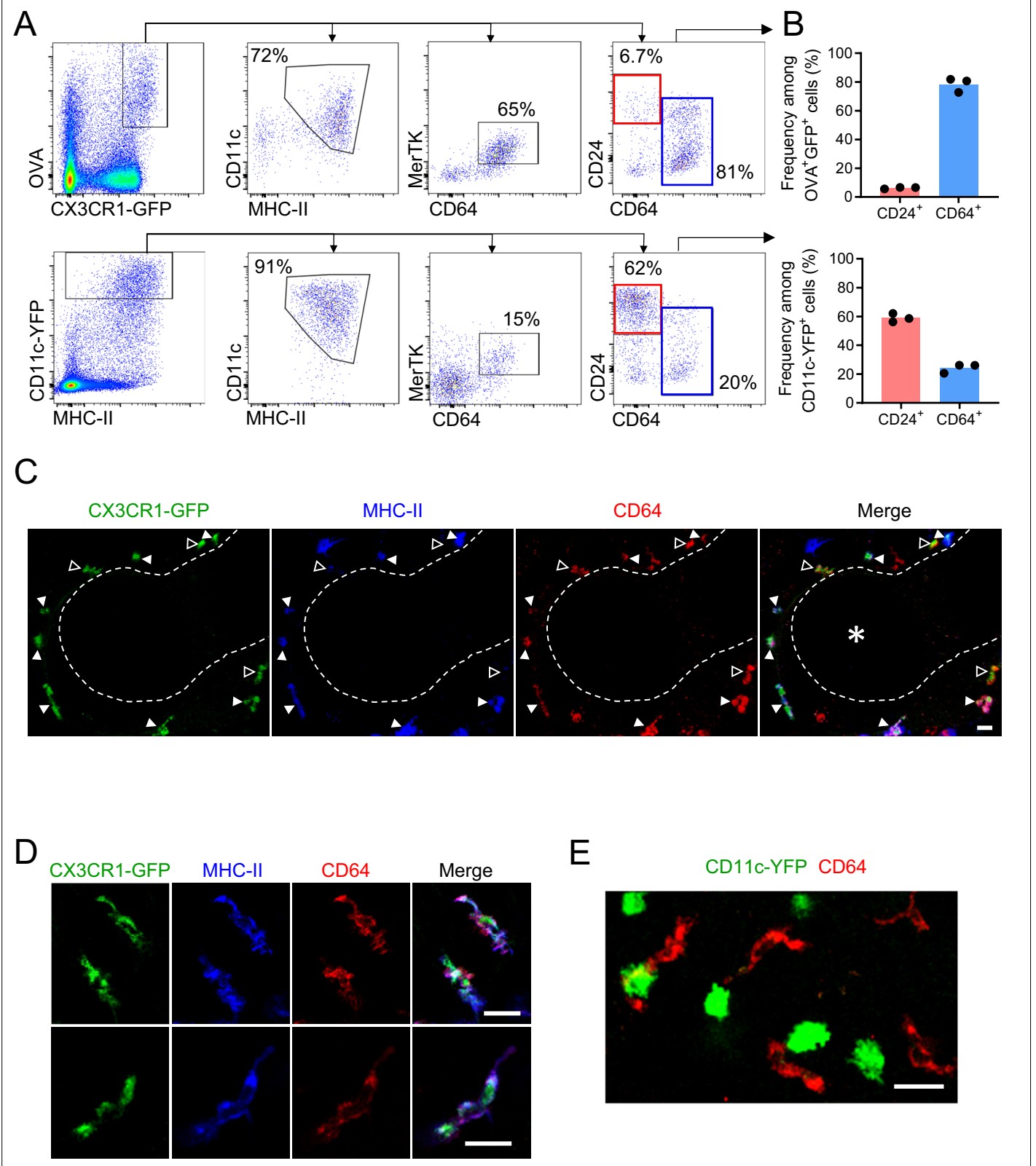

**Figure 3.** Antigen-capturing CX3CR1-GFP<sup>bright</sup> cells are Bronchus-Associated Macrophages (BAMs). (**A**) Representative flow cytometric analysis of CX3CR1-GFP<sup>bright</sup> OVA<sup>bright</sup> cells (top) versus CD11c-YFP<sup>bright</sup> cells (bottom). Cells were pre-gated to exclude B cells, neutrophils, eosinophils, and AMs (CD19⁻ Ly6G⁻ Siglec-F⁻). (**B**) Quantification of the frequency of CD24⁺ CD64⁻ ("CD24⁺") or CD64⁺ CD24⁻/lo ("CD64⁺") cells among CX3CR1-GFP<sup>bright</sup> OVA<sup>bright</sup> cells (top) or CD11c-YFP<sup>bright</sup> cells (bottom) that were gated as in (**A**). (**C–D**) Representative two-photon microscopy of MHC-II (blue) and

*Figure 3 continued on next page*

*Figure 3 continued*

CD64 (red) antibody staining in precision-cut vibratome lung sections from CX3CR1-GFP (green) mice, showing a region containing an airway as a 3D fluorescence opacity rendering of a 54 µm z-stack (**C**) as well as single z-plane images of cells to confirm colocalization of molecules (**D**); note **C** and **D** are from different imaging fields. In (**C and D**), single color and merged images are shown. In (**C**), the bronchial airway was visualized by transmitted illumination and the approximate boundaries are denoted by dashed lines, with the airway lumen denoted by * in the merged image. In (**C**), most CX3CR1-GFP^bright CD64^+ cells are MHC-II^hi (solid arrowheads), yet some cells are MHC-II^lo (open arrowheads). (**E**) Representative two-photon microscopy of CD64 antibody staining (red) in vibratome lung sections from CD11c-YFP (green) mice, showing a MIP of a 15 µm z-stack. Mice in all panels were unimmunized. Each data point represents one mouse (**B**). Data in all panels are representative of ≥2 experiments. Scale bars, 20 µm.

The online version of this article includes the following figure supplement(s) for figure 3:

**Figure supplement 1.** CD11c antibody labeling, CD11c-YFP and CX3CR1-GFP expression on CD64^+ IMs.

**Figure supplement 2.** Phenotypic comparison of cDCs, BAMs and AMs.

suggested they were not derived from inflammatory monocytes. The CX3CR1-GFP^bright OVA^bright cells also predominantly expressed the macrophage markers CD64 and MerTK, and had relatively low to undetectable staining for the cDC marker CD24 (*Figure 3A and B*), consistent with surface markers previously established for IMs (*Gautier et al., 2012*; *Guilliams et al., 2013*; *Schlitzer et al., 2013*). Conversely, the majority of the CD11c-YFP^bright cells were MHC-II^+, CD11c^mid-hi and CD24^hi but did not express CD64 and MerTK, thus representing cDCs (*Figure 3A and B*). Notably, CD64^+ OVA^bright cells were largely CD11c-YFP^− but did express surface CD11c by antibody staining (*Figure 3—figure supplement 1A*), albeit with reduced abundance compared to cDCs, indicating the expression of the CD11c-YFP transgene is more restricted to DCs than the CD11c protein, consistent with a prior study (*Rodero et al., 2015*). The small fraction of CD64^+ OVA^bright cells that were also CD11c-YFP^bright (*Figure 3—figure supplement 1A*) may correspond to the rare cells we had observed by microscopy that were bright for both CX3CR1-GFP and CD11c-YFP (*Figure 2—figure supplement 2B*). We found that monocytes, which resolve into two subsets based on CX3CR1-GFP expression (*Geissmann et al., 2003*), and the CX3CR1-GFP^+ subset of cDC2s, all had lower CX3CR1-GFP expression than the MHC-II^+ CD64^+ MerTK^+ cells (*Figure 3—figure supplement 1B*), consistent with our imaging data (*Figure 2—figure supplement 2A*). Again, as microscopy has a more limited dynamic range than flow cytometry and we had focused our imaging on bright cells using limiting laser power, this likely favored our specific visualization of CX3CR1-GFP^bright cells. To indeed verify that the CX3CR1-GFP^bright cells we had observed near the bronchial airway epithelium by microscopy were represented by the CX3CR1-GFP^bright CD64^+ MHC-II^+ IMs identified by flow cytometry, we stained precision-cut thick lung slices from CX3CR1-GFP or CD11c-YFP mice with antibodies against CD64 and MHC-II. By two-photon microscopy, we confirmed that the CX3CR1-GFP^bright cells located near the bronchial airways were CD64^+ (*Figure 3C and D*). In contrast, the CD11c-YFP^bright cells were CD64-negative (*Figure 3E*). Most of the CX3CR1-GFP^bright cells located near the airways were MHC-II^+ (*Figure 3C* solid arrowheads and *Figure 3D*), although occasional MHC-II^lo/− cells were also visualized (*Figure 3C*, open arrowheads). Taken together, these findings indicate that the majority of the CX3CR1-GFP^bright cells localized near the bronchial airway epithelium that capture soluble inhaled antigens are CD11c^+ CD64^+ MHC-II^+ IMs. Based on their spatial localization near the bronchial airways, we propose that this subset of IMs be referred to as Bronchus-Associated Macrophages (BAMs).

We further characterized BAMs in an immunophenotyping analysis by flow cytometry comparing CD64^+ MHC-II^+ cells to cDCs and AMs. CD64^+ MHC-II^+ BAMs had high surface expression of CD11b and were uniformly MerTK^+, but had only minimal expression of a Zbtb46-GFP reporter that was nearly 10-fold lower than CD24^+ DCs (*Figure 3—figure supplement 2A-C*), further confirming these cells are not DCs despite sharing similar markers to cDC2s. Whereas AMs had high surface expression of CD169 and Siglec-F as expected, we found that CD64^+ MHC-II^+ BAMs had low CD169 expression that partially overlapped with background staining, and did not express Siglec-F (*Figure 3—figure supplement 2D and F*), further supporting that these are two distinct macrophage populations.

To compare gene expression profiles between BAMs and other CD11c^+ cells, particularly DCs and AMs, we further analyzed these populations by single cell RNA sequencing (scRNAseq) analysis, using a multi-step protocol we devised to enrich for CD11c^+ cells from the lungs of unimmunized mice (see Methods for details). Given the large abundance of CD11c^+ AMs in the lung, we first depleted some of these cells to help enrich for IM and DC subsets. This allowed us to sequence hundreds of IMs and DCs from each sample. A substantial number of AMs still remained, allowing us to do a comparison of IMs,

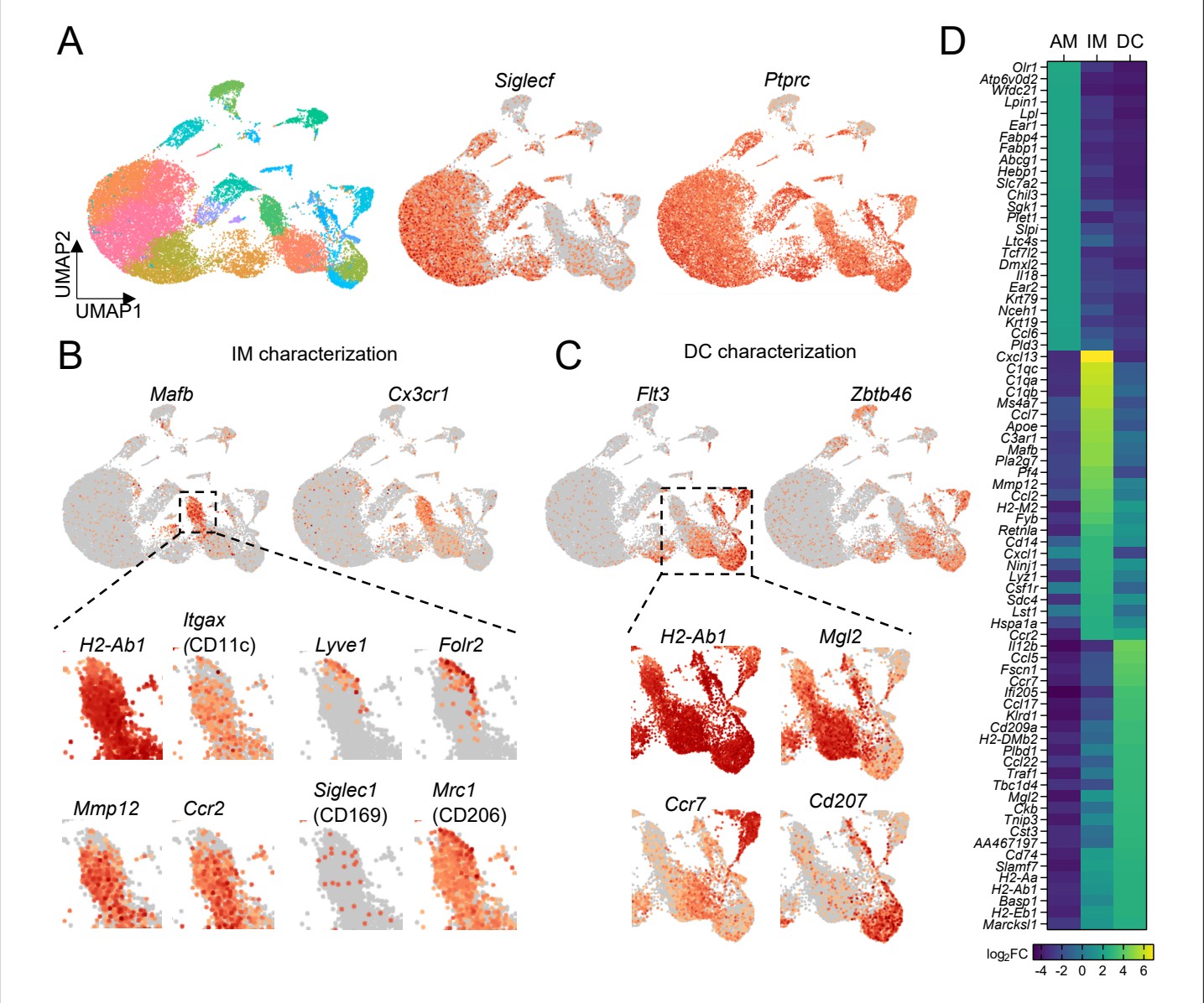

**Figure 4.** Comparison of transcriptional profiles of AMs, IMs, and DCs by scRNA-seq. (**A**) UMAP plot depicting automated graph-based clustering of CD11c-enriched cells aggregated from lungs of four naïve wild-type mice. The abundance of *Siglecf* and *Ptprc* (CD45) transcripts are shown in middle and right plots. (**B**) Transcriptional profile of IMs identified as the cluster of cells most highly expressing *Mafb* and *Cx3cr1*. Genes of interest that were previously reported to be expressed in IMs are shown in the bottom panels. (**C**) Transcriptional profile of DCs identified as the clusters of cells most highly expressing *Flt3* and *Zbtb46*. Specific transcripts for each of the sub-clusters are shown in the bottom panels. (**D**) Heat map showing the top 25 most upregulated, differentially-expressed genes from each of the AMs, IMs, and DCs populations gated according to *Figure 4—figure supplement 1*.

The online version of this article includes the following figure supplement(s) for figure 4:

**Figure supplement 1.** Gating for AMs, IMs, and DCs for scRNA-seq.

DCs, and AMs. Most of the cells sequenced expressed *Ptprc*, the gene encoding CD45, confirming most of the cells analyzed were leukocytes, with minor populations of structural cells (*Figure 4A*). Unsupervised clustering of cells visualized by UMAP revealed multiple clusters that expressed *Siglecf*, with one predominant grouping of clusters that we defined as AMs (*Figure 4A* and *Figure 4—figure supplement 1*). One major cluster of IMs was detected, marked by the highest density of expression of *Cx3cr1*, consistent with our imaging analysis (*Figure 4B* and *Figure 4—figure supplement 1*). The IM cluster also had high expression of *Mafb* (*Figure 4B*), which is thought to largely be restricted to

cells of the macrophage lineage (*Wu et al., 2016*), with the exception of AMs (*Gautier et al., 2012*; *Gibbings et al., 2017*). Within the major IM cluster two distinct populations were visible. BAMs make up the majority of the population of IMs, which expressed comparable amounts of *H2-Ab1* (MHC-II) to the DC subsets, consistent with our flow cytometric analysis, as well as *Itgax* (CD11c), *Mmp12* and *Ccr2* (*Figure 4B*). A minor population of IMs was visible with high expression of *Lyve1* and *Folr2*, with weaker expression of *H2-Ab1* and *Itgax* (*Figure 4B*), consistent with previous reports (*Chakarov et al., 2019*; *Schyns et al., 2019*; *Ural et al., 2020*). *Mrc1* (CD206) was widely expressed among IMs, with the highest expression on the *Lyve1*+ population (*Figure 4B*). *Siglec1* (CD169) was only detected in a small fraction of cells in the IM cluster (*Figure 4B*), suggesting reads were near the threshold for detection, consistent with our CD169 staining by flow cytometry (*Figure 3—figure supplement 2D*). Several clusters of DCs were identified by a high density of expression of *Flt3* and *Zbtb46* (*Figure 4C*). These clusters showed differential expression of *Mgl2*, *Cd207*, and *Ccr7*, suggesting they represented cDC2s, cDC1s, and migratory DCs, respectively (*Izumi et al., 2021*; *Sichien et al., 2017*). We grouped these DC clusters together for comparison to AMs and IMs (*Figure 4—figure supplement 1*). Several genes were identified that were most highly expressed in IMs compared to AMs and DCs: these included genes encoding chemokines (*Cxcl13*, *Ccl7*, and *Ccl2*), complement pathway components (*C1q* genes and *C3ar1*), *Apoe*, and *Mmp12* (*Figure 4D*). This transcriptional profile analysis demonstrates that despite similarities in their surface marker profiles, BAMs, AMs, and DCs are functionally distinct subsets of cells.

Previous studies have reported lung IMs can be divided into subsets based on CD206 expression (*Chakarov et al., 2019*; *Gibbings et al., 2017*; *Schyns et al., 2019*). One of these studies reported that airway associated IMs were largely MHC-II⁻ and CD206⁺, whereas MHC-II⁺ CD206⁻ IMs were predominantly located near the alveolar parenchyma (*Schyns et al., 2019*). Our microscopy data above, however, had clearly demonstrated CX3CR1-GFP^bright MHC-II⁺ IMs were positioned near the airways. Our scRNAseq data above further indicated that MHC-II⁺ IMs express *Mrc1*, the gene encoding CD206, although *Mrc1* was most highly expressed in the *Lyve1*+ subset of IMs. Our flow cytometric analysis of CD64⁺ MHC-II⁺ cells showed that BAMs had intermediate CD206 surface expression in a Gaussian distribution (*Figure 3—figure supplement 2E*). By two-photon microscopy of precision-cut thick lung slices, we observed that the CX3CR1-GFP^bright BAMs that predominantly captured antigen (OVA) near the airways were CD206⁺ (*Figure 5A*), although the expression of CD206 was variable. By co-staining with MHC-II, we observed that CX3CR1-GFP^bright MHC-II^hi cells near the airways (*Figure 5B*, solid arrowheads) expressed variable intermediate amounts of CD206, whereas the highest CD206 expression was observed on some CX3CR1-GFP⁺MHC-II^lo cells that were also located near the airways (*Figure 5B*, open arrowheads). Taken together, these data establish that BAMs, which capture antigen near the airways and are MHC-II⁺, express variable amounts of CD206.

Two recent studies reported that some IMs in the lung were located near nerves (*Chakarov et al., 2019*; *Ural et al., 2020*), although this was less apparent in another study (*Schyns et al., 2019*). Notably, bronchovascular bundles contain bronchi, arteries, lymphatic vessels, and nerves. By two-photon microscopy of precision-cut thick lung slices stained with an antibody to L1CAM, a neuronal cell adhesion molecule (*Rathjen and Schachner, 1984*; *Tacke et al., 1987*), we readily visualized nerves near the bronchial airways and at airway branchpoints where BAMs were abundant (*Figure 5C and D*). A subset of BAMs were directly in contact with nerves, while other BAMs were not in contact with nerves (*Figure 5C and D*). One recent study defined a subset of IMs as nerve and airway associated macrophages (NAMs) (*Ural et al., 2020*). However, this population of NAMs was notable for very high CD169 expression, yet our flow cytometric and scRNAseq analyses suggested the antigen-capturing BAMs near the airways were low to negative for CD169 expression (*Figure 3—figure supplement 2D* and *Figure 4B*). In two-photon microscopy of precision-cut thick lung slices, we observed punctate CD169 staining on BAMs that was only detectable above background in a subset of cells (*Figure 5D*, solid arrowheads). In contrast, we occasionally observed cells with bright uniform CD169 staining in contact with nerves; however, these cells were not CX3CR1-GFP^bright and thus were distinct from BAMs (*Figure 5D*, open arrowhead). Interestingly, the punctate CD169 staining on BAMs was polarized toward one side of the cell, and this often seemed to be at the interface with a CD11c-YFP⁺ DC (*Figure 5E*). A previous study demonstrated an important role for CD169 in cell-cell contacts between macrophages and DCs in the spleen and lymph nodes, which enhanced antigen transfer and cross-presentation (*van Dinther et al., 2018*). In our static imaging data above, we noted that

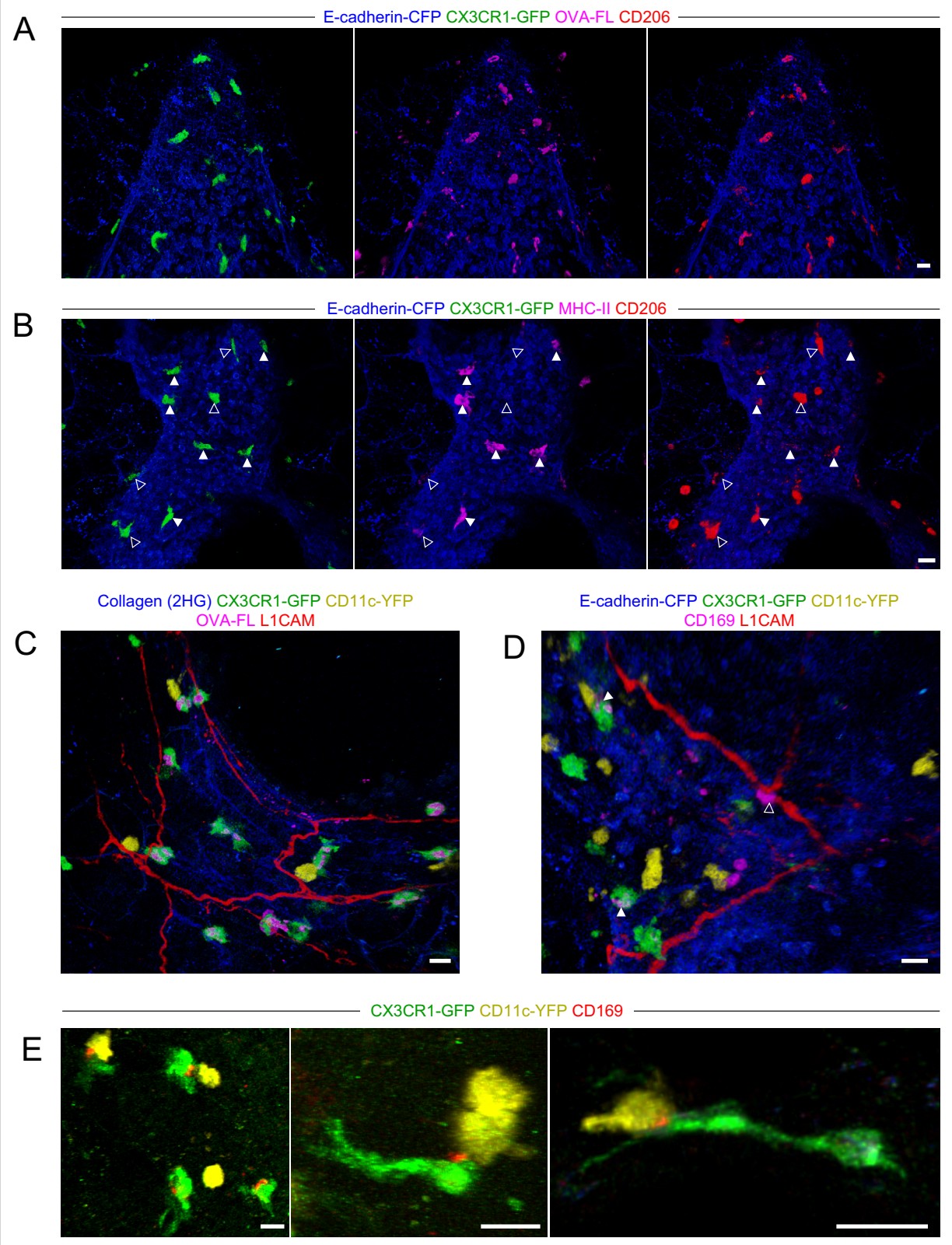

**Figure 5.** Characterization of BAM marker expression and positioning with respect to nerves. Lungs were isolated from naïve mice expressing E-cadherin-CFP (blue), CX3CR1-GFP (green), and/or CD11c-YFP (yellow), as indicated. Some mice received fluorescent OVA (OVA-FL) 2 hr prior to imaging. Precision-cut lung slices were labeled with the indicated antibodies (see Methods for details) and then imaged by two-photon microscopy. Shown are 3D fluorescence opacity renderings of image stacks of regions adjacent to airways. (**A, B**) Comparison of CD206 staining (red) to OVA-FL

*Figure 5 continued on next page*

*Figure 5 continued*

uptake (magenta, **A**) and MHC-II staining (magenta, **B**). Separate two-color images (each color with E-cadherin-CFP) are shown for best visualization of the different fluorophores. In (**B**), CX3CR1-GFP^bright MHC-II^hi cells are indicated by solid arrowheads and CX3CR1-GFP^bright MHC-II^lo cells are indicated by open arrowheads. (**C, D**) Visualization of nerves with L1CAM staining (red) together with OVA-FL uptake (magenta, **C**) and CD169 staining (magenta, **D**). Merged color images are shown to enable clear visualization of spatial localization of cells with respect to nerves. In (**D**), punctate CD169 staining on BAMs is indicated by solid arrowheads, whereas a CX3CR1-GFP^neg cell with uniform, bright CD169 staining adjacent to the nerve is shown with an open arrowhead. (**E**) Examples of punctate CD169 staining (red) on BAMs, often at the interface between BAMs and DCs. Images were derived from z-stacks of 182 μm (**A**), 102 μm (**B**), 127 μm (**C**), 219 μm (**D**), 165 μm (**E**, left), 124 μm (**E**, middle), and 7.5 μm (**E**, right). Scale bars, 20 μm.

BAMs and DCs were frequently in contact (see *Figure 2D and F*, *Figure 2—figure supplement 2*, and *Figure 3E*). We observed extensive, dynamic interactions between BAMs and DCs in time-lapse imaging of precision-cut lung slices by two-photon microscopy (*Video 4*). This suggests the possibility that BAMs may transfer antigen, or peptide-MHC II complexes, to DCs. Taken together, in addition to their role in antigen capture from the airways, BAMs may interact with nerves and/or communicate with DCs.

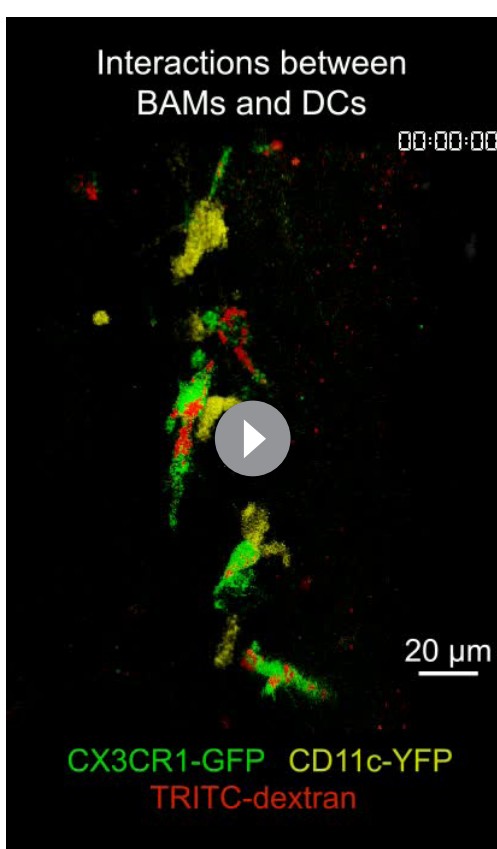

**Video 4.** Antigen-capturing BAMs interact with DCs. The video is a time-lapse image sequence showing the interactions of CX3CR1-GFP^bright BAMs (green) with CD11c-YFP^bright DCs (yellow). The BAMs appear to have captured 10 kDa TRITC-dextran (red), which was administered to the mouse by intranasal droplet approximately 2 hr prior to lung isolation. The image sequence is rendered from a 32 μm z-stack collected by two-photon microscopy of a cultured precision-cut vibratome section of the lung. Elapsed time is shown as hh:mm:ss.

https://elifesciences.org/articles/63296/figures#video4

## BAMs primarily capture lower molecular weight soluble proteins

We next further characterized the ability of BAMs to take up antigens from the airway lumen. To compare the antigen uptake capacities of BAMs to individual subsets of DCs and monocytes, fluorescent OVA was administered to mice intranasally 2 hr before lungs were digested and dissociated into single-cell suspensions for flow cytometric analysis. We gated on CD45^+ CD19^− Ly6G^− Siglec-F^− CD11c^+ MHC-II^+ CD11b^+ CD24^{−/lo} CD64^+ cells as a reasonable representation of BAMs, as we had established based on our marker analysis above. Indeed, a large majority (~80%) of BAMs gated by these markers had taken up OVA, whereas a smaller proportion (~50%) of cDC2s and less than 5% of monocytes were OVA^+ (*Figure 6A*). By microscopy, we observed that in addition to soluble OVA and HDM extract, CX3CR1-GFP^+ BAMs took up other soluble fluorescent molecules of a range of sizes, such as TRITC-dextran molecules that were 10 kDa or 155 kDa, as well as phycoerythrin (PE), a 240 kDa molecule (*Figure 6B–D*). To quantify whether antigen size affected the efficiency of antigen capture by BAMs, we administered FITC-dextran (FITC-DX) of various sizes ranging from 10 kDa to 2 MDa, and then assessed the frequency of FITC-DX^+ cells among BAMs (*Figure 6E*, left) and cDC2s (*Figure 6E*, right) by flow cytometry. To exclude the possibility that FITC-DX was taken up during the lung digestion process, lungs from congenic mice that were not exposed to FITC-DX (control lung) were processed in the same digestion reaction as the lungs from the mice that had received FITC-DX (host lung). Very few (or less than 5% of) BAMs from the control lungs of congenic mice that were co-digested were FITC-DX^+, confirming that minimal FITC-DX uptake occurred during the

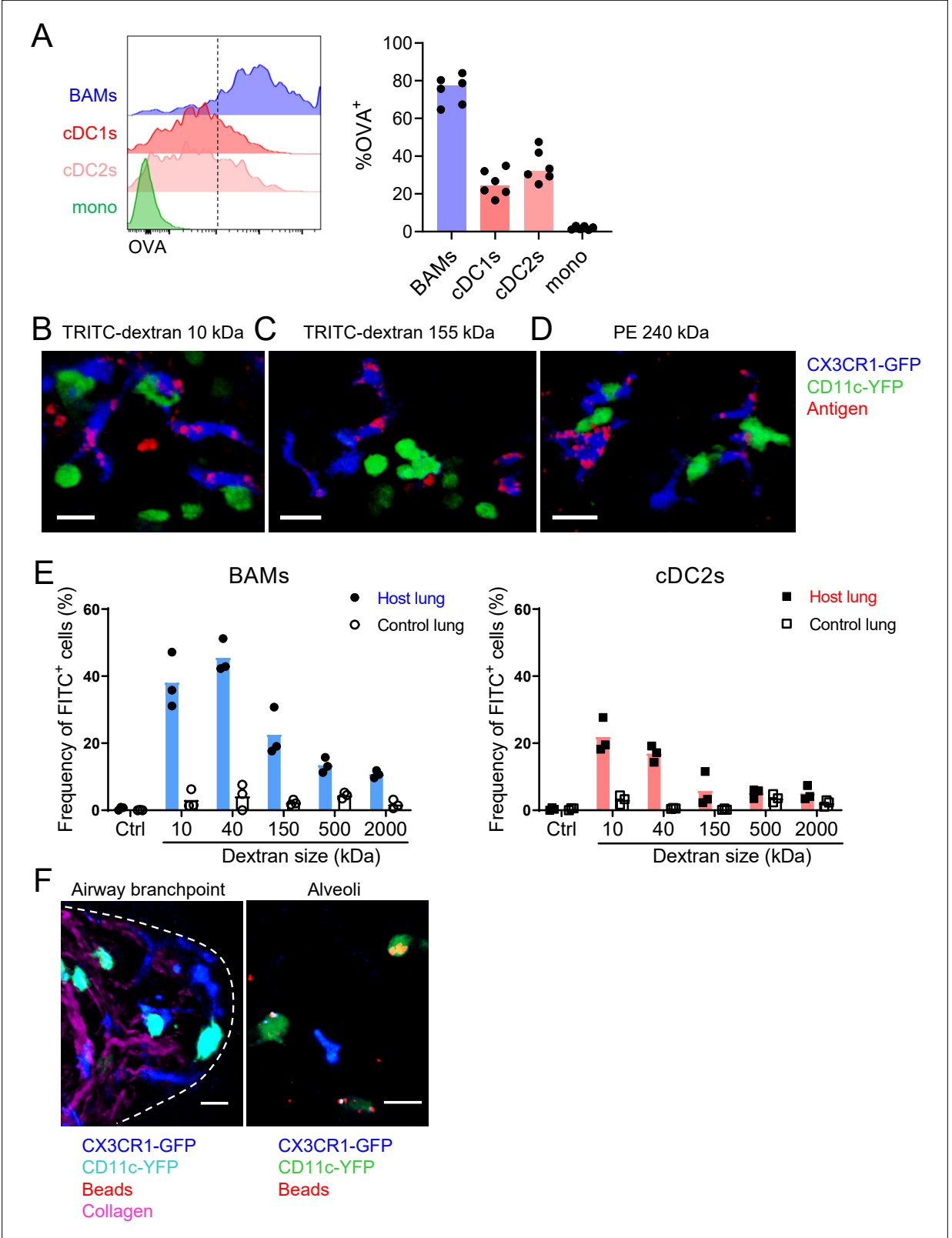

**Figure 6.** Characteristics of molecules captured by BAMs versus cDCs. (**A**) Fluorescent OVA was administered by intranasal droplet to naïve mice 2 hr before lung isolation and enzymatic digestion. Representative flow cytometry (left) and quantification (right) of the frequency of OVA$^+$ cells within each indicated cell subset. Cells were pre-gated to exclude B cells, neutrophils, eosinophils, and AMs (CD19$^-$ Ly6G$^-$ Siglec-F$^-$). CD11c$^+$ MHC-II$^+$ cells were resolved into CD24$^+$ cDCs versus CD64$^+$ BAMs as in *Figure 4A*, and then the remaining cells that were not CD11c$^+$ MHC-II$^+$ cells were gated as CD11b$^+$

*Figure 6 continued on next page*

*Figure 6 continued*

to define monocytes (mono). cDC1s were gated as CD11b⁻ CD103⁺ and cDC2s were gated as CD11b⁺ CD103⁻ subsets of cDCs. Mice were either CX3CR1-GFP or CD11c-YFP mice. In CX3CR1-GFP mice, monocytes appeared as two subsets with low to intermediate GFP expression, consistent with classical and non-classical monocytes, respectively (data not shown). (**B, C, D**) The indicated fluorescent antigens (red) were administered by intranasal droplet to naïve CX3CR1-GFP (blue) X CD11c-YFP (green) mice 2–4 hr before lung isolation and the preparation of precision-cut thick lung sections. Representative images depict cells that captured the indicated fluorescent molecules in 3D fluorescence opacity renderings of 73 μm (**B**), 16 μm (**C**), and 51 μm (**D**) z-stacks collected by two-photon microscopy. DX, dextran; scale bars, 20 μm. (**E**) FITC-dextran of a range of sizes was administered by intranasal droplet to naïve mice, and then 2–3 hr later the lungs were isolated and enzymatically digested. The frequencies of FITC⁺ BAMs (left) and FITC⁺ cDC2s (right) were enumerated by flow cytometry as shown. To control for the possibility of FITC-dextran uptake during lung digestion and dissociation, the lungs from congenic mice that did not receive FITC-dextran were co-digested ('control lung') with the lungs of mice that had received FITC-dextran ('host lung'). (**F**) Fluorescent beads (red) were administered to naïve mice expressing CX3CR1-GFP (blue) and CD11c-YFP (cyan, left or green, right). Representative two-photon microscopy images of an airway branchpoint (left) and alveoli (right) are shown. Images are MIPs of 107 μm (left) and 45 μm (right) z-stacks. Beads were primarily captured by AMs which appear dimly fluorescent in the YFP channel (note that the image of the alveoli [right] is adjusted to show the dimmer signal). Collagen (magenta) was visualized by 2HG. Each data point represents one mouse (**A, E**). Data are representative of two experiments (**A–D**). Similar results to (**F**) were observed in other experiments with smaller and larger beads. Scale bars, 20 μm.

digestion procedure (*Figure 6E*). We found that a greater proportion of BAMs were FITC-DX⁺ when smaller molecules were instilled compared to larger molecules (*Figure 6E*). Therefore, the magnitude of antigen uptake by BAMs is inversely related to the antigen size.

It has been reported that DCs in the alveolar interstitium send projections into the alveolar space to capture fluorescent bead particles (*Thornton et al., 2012*). To assess whether BAMs captured particulate antigens from the bronchial airway lumen, we intranasally instilled fluorescent beads and analyzed the lungs by microscopy. However, we rarely observed bead capture by CX3CR1-GFP⁺ BAMs or CD11c-YFP⁺ DCs near the airways (*Figure 6F*). In contrast, in the alveolar space, beads were taken up primarily by highly spherical cells that were dimly CD11c-YFP⁺ (*Figure 6F*). By location and morphology, we concluded that these cells were AMs, not DCs. In summary, BAMs preferentially captured soluble antigens, whereas both BAMs and DCs residing at the airway interstitium showed minimal capture of beads.

## BAMs activate local effector T cells

Based on our findings that BAMs readily captured soluble antigens from the bronchial airways and had high surface expression of MHC-II comparable to DCs, we considered whether BAMs are capable of activating T cells at these sites. In order to activate T cells, APCs need to process antigen and present antigen peptides on MHC-II on their cell surface. We used the Y-Ae antibody to test whether BAMs were able to process and present antigen. The Y-Ae antibody specifically recognizes the Eα peptide, a peptide fragment of the I-E MHC-II molecule, presented on the I-Aᵇ MHC-II molecule (*Murphy et al., 1992*). C57BL/6 mice do not express I-E and thus Eα represents a foreign peptide. The Eα peptide was fused to OVA, administered intranasally together with fluorescent OVA to label antigen-capturing cells, and then OVA⁺ lung APCs were analyzed 17–18 hr later to allow time for antigen processing and presentation. Both BAMs and cDC2s stained with the Y-Ae antibody by flow cytometry (*Figure 7A*), demonstrating that BAMs processed and presented antigen, like DCs, rather than simply degrading it. Minimal Y-Ae staining was observed 15 min after OVA-Eα was administered, confirming that antigen processing and presentation were needed for detection of MHC-II:Eα complexes on the cell surface (data not shown). Brighter Y-Ae staining was observed on BAMs compared with cDC2s (*Figure 7A*), consistent with the superior soluble antigen uptake we had observed (*Figures 2 and 6*). AMs, with low MHC-II expression and a high degradative capacity, showed negligible Y-Ae staining (*Figure 7A*), confirming these are not the relevant APCs for CD4 T cells. We also tested whether BAMs could provide the necessary co-stimulatory signals for T cell activation and found that BAMs had higher surface expression of the co-stimulatory molecules CD80 and CD86 compared to cDC2s (*Figure 7B*). To determine whether BAMs were able to activate CD4 T cells, BAMs were sorted by FACS and co-cultured with OT-II TCR transgenic T cells, which respond to an OVA peptide presented on I-Aᵇ by APCs. As a positive control for APC function in cell culture, cDC2s were sorted and co-cultured with OT-II T cells. When cultured in the presence of OVA peptide, BAMs induced the robust proliferation of naïve OT-II T cells, albeit somewhat less efficiently than cDC2s, as expected, given the high efficiency of naïve T cell activation by DCs (*Figure 7C*). Interestingly, BAMs and cDC2s equivalently induced IL-13 production when co-cultured with Th2 polarized, effector OT-II cells and OVA peptide

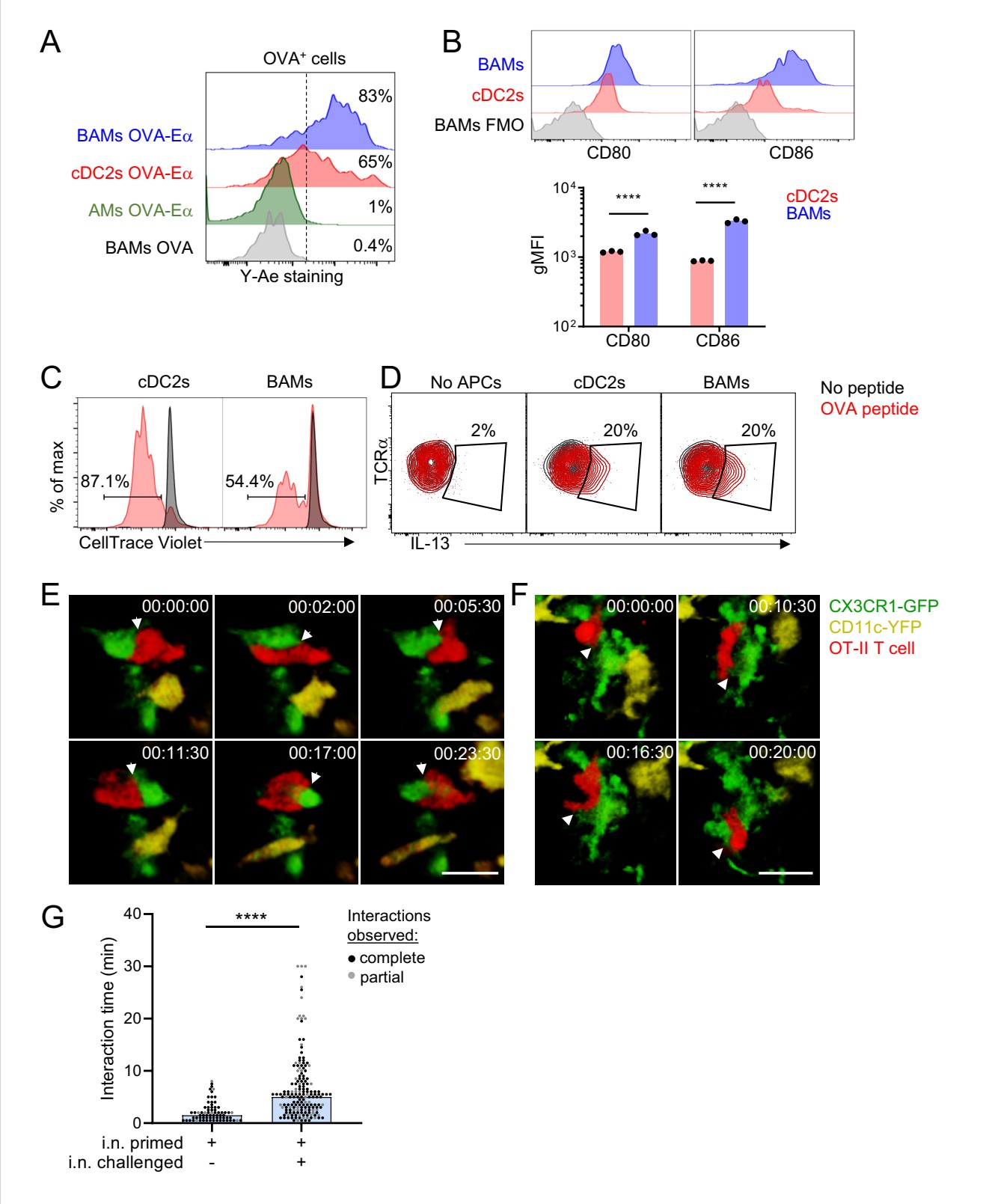

**Figure 7.** BAMs act as APCs and interact with CD4 effector T cells in the lung. (**A**) Representative flow cytometric analysis of Y-Ae antibody staining to detect the presentation of MHC-II:Eα peptide complexes. OVA conjugated to Eα peptide was administered intranasally together with OVA-Alexa647 to fluorescently label antigen-capturing cells, and then after 17–18 hr to allow time for antigen processing and presentation, the lungs were isolated, followed by enzymatic digestion and analysis of Alexa647+ BAMs (blue), cDC2s (red) and AMs (green). Control mice received OVA without the Eα

*Figure 7 continued on next page*

*Figure 7 continued*

peptide (gray). (**B**) Representative flow cytometry (top) and quantification (bottom) of the surface expression of the co-stimulatory molecules CD80 and CD86 on BAMs (blue) and cDC2s (red) in the lungs of naïve mice in the steady state. FMO (gray), fluorescence minus one control. (**C**) Flow cytometric analysis of the proliferation of naïve OT-II T cells that were co-cultured for 2.5 days with sorted cDC2s or BAMs in the presence (red) or absence (black) of OVA peptide. Proliferation was assessed by dilution of CellTrace Violet. (**D**) Flow cytometric analysis of IL-13 production in Th2-polarized OT-II T cells that were cultured overnight with sorted cDC2s or BAMs, or no APCs as a control, in the presence (red) or absence (black) of OVA peptide, together with Brefeldin A. (**E–G**) Time-lapse images (**E, F**) and quantification (**G**) of CD4 T cell interactions with CX3CR1-GFP⁺ BAMs. OT-II T cells (red) expressing tdTomato (**E**) or CFP (**F**) were adoptively transferred to mice expressing CX3CR1-GFP (green) and CD11c-YFP (yellow) that had been sensitized with OVA and low-dose LPS intranasally, and then challenged with OVA alone intranasally 9–10 days later. Precision-cut thick vibratome sections of the lung were prepared for imaging 4–6 hr after OVA challenge, and two-photon microscopy z-stacks of 15 μm (**E**) and 22 μm (**F**) are shown. (**G**) Quantification of interaction times in which CX3CR1-GFP⁺ BAMs were in direct contact with OT-II T cells. Each data point represents a BAM-T cell interaction event. Due to the limitations of imaging in a specific volume over a fixed period of time, only some cellular interactions could be tracked for their entire duration from start to finish (complete, black dots), whereas other interactions could only be tracked for a portion of the total interaction time (partial, gray dots) and therefore are underestimates. Elapsed time is indicated as hh:mm:ss. Each data point represents one mouse (**B**). Data are representative of three experiments (**A, D**), two experiments (**B, C**), and five experiments (**E–F**). Data in (**G**) are pooled from three time-lapse imaging sessions from three separate experiments. ****p<0.0001 (Mann-Whitney test). Scale bars, 20 μm.

(*Figure 7D*). These findings indicate that BAMs possess the necessary machinery for antigen presentation and activation of T cells, and further suggest that BAMs may equivalently activate effector T cells compared with DCs.

We hypothesized that the primary function of MHC-II expression on BAMs might be to activate local effector/memory T cells that have been recruited to the lung tissue. To test this hypothesis, we utilized time-lapse two-photon microscopy to determine whether BAMs encountered and interacted with effector T cells in the lungs. Naïve OT-II T cells expressing CFP (β-actin CFP) or tdTomato (CD4-Cre Ai14) were adoptively transferred into CX3CR1-GFP X CD11c-YFP mice and then primed in vivo, which resulted in expansion and recruitment to the lungs. Mice were subsequently rechallenged with intranasal OVA alone, lungs were excised 4 hr later, and precision-cut lung slices were prepared for imaging. After allergen challenge, we observed sustained interactions between OT-II T cells and CX3CR1-GFP⁺ BAMs in the lung (*Figure 7E–G*, *Videos 5 and 6*). These interactions between BAMs and OT-II T cells were comparable to the interactions reported during antigen-specific T cell activation by DCs in the lymph nodes (*Mempel et al., 2004*). Some OT-II T cells were also observed to engage in sustained interactions with CD11c-YFP⁺ DCs near the airways (data not shown). These observations suggest that both BAMs and DCs can serve as APCs for the local activation of effector T cells in lung peribronchial regions.

## BAMs are non-migratory while DCs are migratory

Although BAMs and DCs both exhibited the capacity to serve as APCs for T cells, time-lapse imaging by two-photon microscopy revealed fundamental differences in cellular dynamics. BAMs had dendrites that were continuously

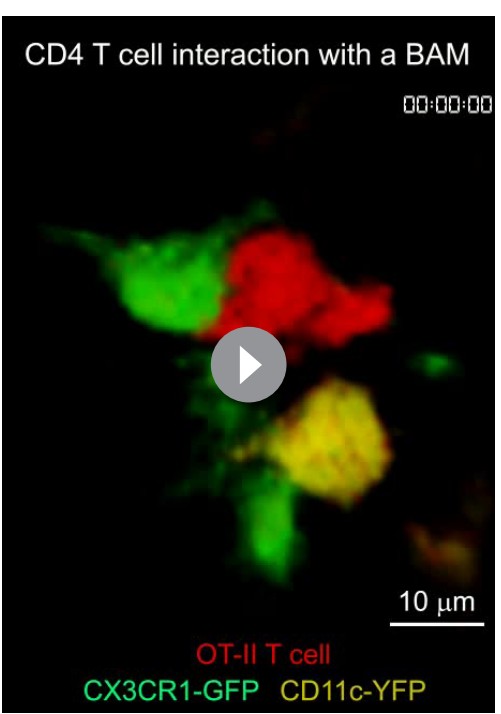

**Video 5.** CD4 T cell interaction with a BAM (example 1). The video is a time-lapse image sequence corresponding to Figure 6E, showing the interactions of an OVA-specific OT-II CD4 T cell (red) with a CX3CR1-GFPᵇʳⁱᵍʰᵗ BAM (green) in the proximity of CD11c-YFPᵇʳⁱᵍʰᵗ DCs (yellow). The OT-II T cell was visualized by the expression of CD4-Cre and the Ai14 reporter leading to tdTomato expression. The image sequence is rendered from a 15 μm z-stack, collected by two-photon microscopy of a cultured precision-cut vibratome section of the lung isolated 4 hr after intranasal OVA challenge. Elapsed time is shown as hh:mm:ss. https://elifesciences.org/articles/63296/figures#video5

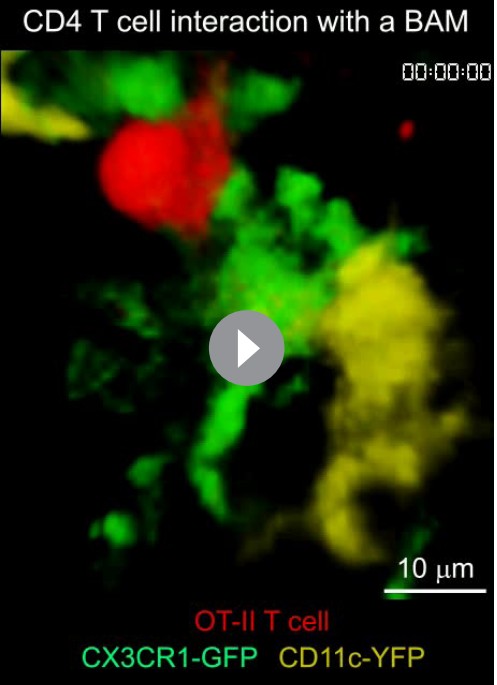

**Video 6.** CD4 T cell interaction with a BAM (example 2). The video is a time-lapse image sequence corresponding to Figure 6F, showing the interactions of an OVA-specific OT-II CD4 T cell (red) with a CX3CR1-GFP^bright BAM (green) in the proximity of CD11c-YFP^bright DCs (yellow). The OT-II T cell was visualized by the expression of a β-actin-CFP transgene. The image sequence is rendered from a 22 μm z-stack, collected by two-photon microscopy of a cultured precision-cut vibratome section of the lung isolated 6 hr after intranasal OVA challenge. Elapsed time is shown as hh:mm:ss.

https://elifesciences.org/articles/63296/figures#video6

extended and probing, but their cell bodies showed very minimal displacement over time. In contrast, DCs were surveying their surroundings by moving their entire cell bodies and traversed greater distances than BAMs (*Figure 8A–D* and *Video 7*). Therefore, DCs were highly migratory whereas BAMs were probing by extending dendrites with little net movement.

Based on our observations of their distinct migratory behavior, we hypothesized that BAMs may remain at the airways, whereas DCs would migrate to the draining lymph node to prime T cells, after antigen capture that leads to an inflammatory response. However, in other studies, cells with similar markers as BAMs, referred to as monocyte-derived DCs, had been observed in the draining lymph node harboring antigen (*Plantinga et al., 2013*). We therefore tested whether BAMs were able to migrate to the right posterior mediastinal lymph node that drains the lung. We first considered the expression of the chemokine receptor CCR7, which is necessary for migration to lymph nodes and is upregulated following exposure to inflammatory stimuli. We analyzed the cell surface expression of CCR7 on DCs and BAMs after administering LPS or HDM as inflammatory stimuli. After 4 hr, we found that CCR7 was upregulated on a significant proportion of DCs in a dose-dependent manner, but not on BAMs (*Figure 8E–F*). We also tested whether OVA, which does not have intrinsic inflammatory properties, would induce CCR7 upregulation in previously sensitized mice in which OVA-specific effector/memory T cells could induce maturation of the APCs. Indeed, after OVA challenge, DCs upregulated CCR7 expression in previously sensitized mice but not in naïve mice exposed to OVA for the first time (*Figure 8G*). Conversely, BAMs

did not upregulate CCR7 in either sensitized or naïve mice (*Figure 8G*). As such, the expression of CCR7 only on the DCs, but not BAMs, would likely allow DCs to selectively migrate to the lymph node. To further assess whether DCs or BAMs entered and migrated in lymphatic vessels within the lung by two-photon microscopy, we visualized lymphatics by breeding Prox1-tdTomato transgenic mice (*Truman et al., 2012*) to CD11c-YFP or CX3CR1-GFP mice. After intranasal instillation of HDM or LPS, we frequently observed CD11c-YFP+ cells, but not CX3CR1-GFP+ cells, migrating within lung lymphatics (*Figure 8H*, *Videos 8 and 9*). To determine which cells carried antigen into the lymph node, we intranasally instilled PE. With a molecular weight of 240 kDa, PE is too large to enter the conduit system of the lymph node (*Roozendaal et al., 2009*) and thus can only reach the lymph node parenchyma when carried by APCs. As an inflammatory stimulus to promote APC trafficking to the lymph node, we administered HDM extract together with the PE. This led to the recruitment of a population of CD64+ MHC-II+ cells that were Ly6C+ to the draining mediastinal lymph node 1 d later. However, these CD64+ MHC-II+ Ly6C+ cells were PE−, whereas we were readily able to identify a fraction of DCs that were PE+ (*Figure 8I–J*). Thus, the CD64+ MHC-II+ Ly6C+ cells were not transporting PE from the lung to the lymph node. The high surface expression of Ly6C on the CD64+ MHC-II+ cells also suggests that they were not BAMs, which have low to moderate surface expression of Ly6C (*Figure 8—figure supplement 1*). Instead, the high expression of Ly6C suggests that the CD64+ MHC-II+ cells were

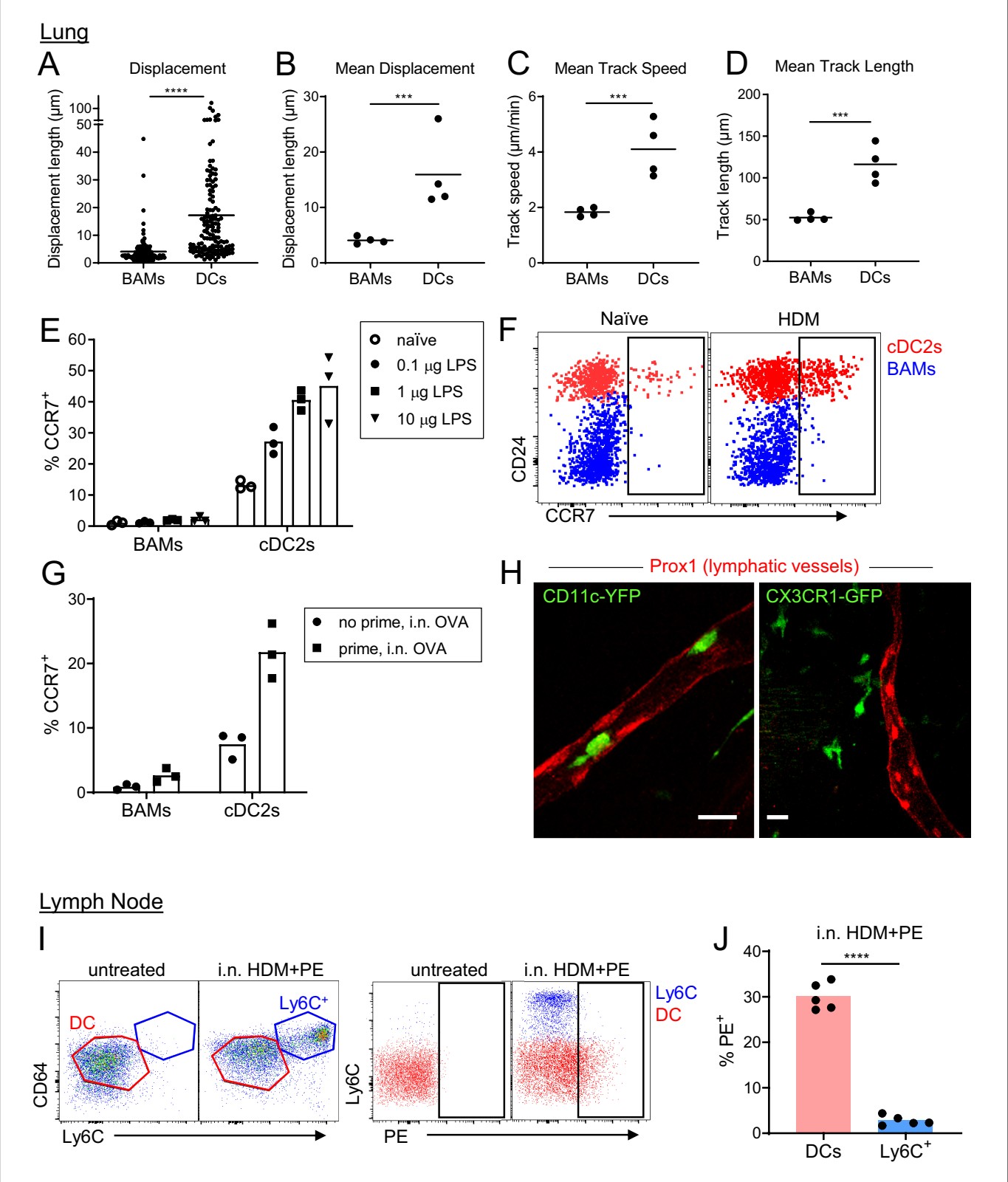

**Figure 8.** BAMs remain lung resident whereas DCs migrate to the draining lymph node. (**A–D**) Motility measurements derived from time-lapse imaging of CX3CR1-GFP+ BAMs and CD11c-YFP+ DCs by two-photon microscopy of precision-cut thick lung sections from naïve mice. (**E**) Flow cytometric analysis of the frequency of CCR7+ cells among BAMs and cDC2s in the lung 4 hr after the indicated concentrations of LPS were administered intranasally. (**F**) Representative flow cytometry showing CCR7 expression on BAMs (blue) and cDC2s (red) in the lung 6 hr after intranasal HDM

*Figure 8 continued on next page*

*Figure 8 continued*

administration. (**G**) The frequency of CCR7⁺ cells among BAMs and cDC2s in the lung in mice that were naïve ('no prime') or had been primed 10 d earlier with intranasal OVA and 100 ng LPS ('prime'), and were challenged once with intranasal OVA (i.n. OVA) without adjuvant. Lungs were excised 6 h after OVA challenge, enzymatically digested, and analyzed by flow cytometry. (**H**) Representative two-photon microscopy images of CD11c-YFP⁺ DCs (green) or CX3CR1-GFP⁺ BAMs (green) and lymphatic vessels (Prox1-tdTomato, red). Mice were administered intranasal LPS 2–4 hr before lung isolation, and then precision-cut thick lung slices were imaged for several hours. Images are MIPs of 41 µm (left) and 32 µm (right) z-stacks. (**I**) Representative flow cytometry gating in the lung-draining mediastinal lymph node in mice that were untreated or had been given HDM and PE intranasally 1 d earlier. Left, gating of cDCs (DC, red) and monocyte-derived cells (Ly6C+, blue). Cells were pregated as CD19⁻ CD11c⁺ MHC-II⁺ CD11b⁺. Right, gating of PE⁺ cells. (**J**) Quantification of the frequency of PE⁺ cells among DCs and monocyte-derived cells (Ly6C⁺) gated as in (**I**) 1 d after HDM and PE were administered intranasally. Each data point represents an individual cell, pooled from four time-lapse image sequences (**A**); the average of all cells from one time-lapse image sequence (**B**, **C**, and **D**); or one mouse (**E**, **G**, **J**). Data are from 2 experiments (**A–D**) or are representative of two experiments (**E, F, I, J**). Similar results to the primed mice shown in (**G**) were observed 1 d after OVA challenge, and similar results for CD11c-YFP cells shown in (**H**) were observed with HDM treatment. *** p<0.001, **** p<0.0001 (t-test). Scale bars, 20 µm.

The online version of this article includes the following figure supplement(s) for figure 8:

**Figure supplement 1.** Comparison of Ly-6C surface expression on BAMs, DCs, and monocytes.

derived from monocytes that were recruited to the lymph node from the bloodstream and had upregulated CD64 and MHC-II in response to the inflammatory conditions. Taken together, these findings indicate that BAMs capture antigen at the airways but do not migrate to the draining lymph node. In contrast, DCs are induced to migrate to the draining LN in response to inflammatory stimuli or in the context of adaptive immune responses.

## Discussion

This work focused on the cell types involved in antigen surveillance at the airways in the steady state to understand the early events that initiate allergic airway inflammation. Our analysis of APCs at the airways revealed an abundant population of MHC-II^hi IMs positioned under the bronchial airway epithelium that sampled soluble luminal antigen, which we term BAMs. We found that BAMs were particularly enriched in collagen-rich regions near some airway branchpoints and engaged in extensive interactions with DCs. In addition, we demonstrated that BAMs were capable of antigen presentation to T cells and interacted with recruited T cells near the airways in an allergic airway disease model. While DCs migrated to the draining lymph node, BAMs remained resident in the lung after antigen capture, even in the presence of inflammatory stimuli. These findings support a model in which BAMs may locally activate effector/memory T cells recruited to the lung, whereas migratory DCs are likely critical for T cell priming in lymph nodes.

In characterizing the localization of lung cDCs and BAMs, we found that in naïve mice, both cell types were positioned around the conducting airways, and were especially enriched at some airway branchpoints in collagen-rich regions. This positioning likely facilitates the uptake of antigens that have deposited at airway branchpoints,

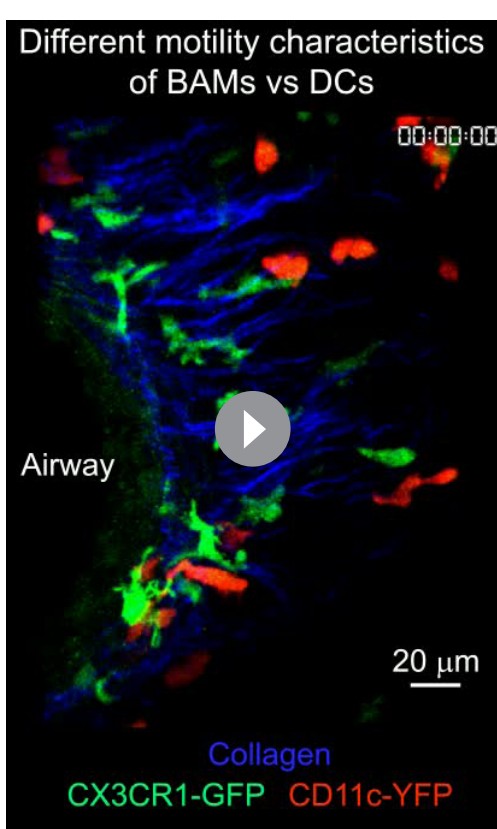

**Video 7.** Different motility characteristics of BAMs vs DCs. The video is a time-lapse image sequence showing the dynamics of CX3CR1-GFP^bright BAMs (green) versus CD11c-YFP^bright DCs (red) in a collagen-rich region proximal to an airway. Collagen (blue) was visualized by second harmonic generation. The MIP represents a 27 µm z-stack collected by two-photon microscopy of a precision-cut vibratome section of the lung of a naïve mouse. Elapsed time is shown as hh:mm:ss.

https://elifesciences.org/articles/63296/figures#video7

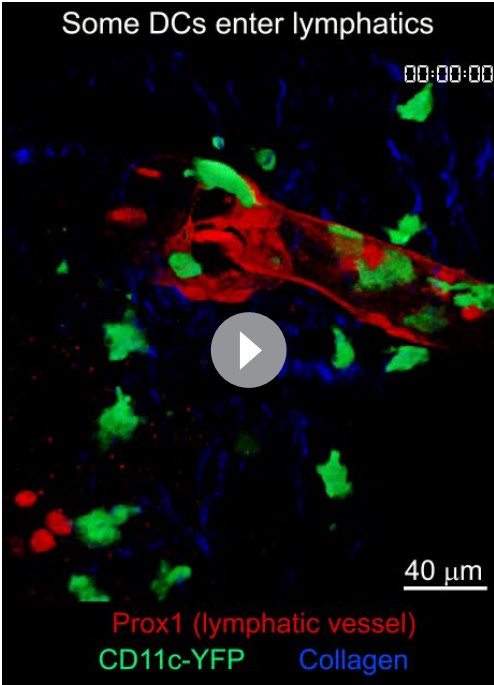

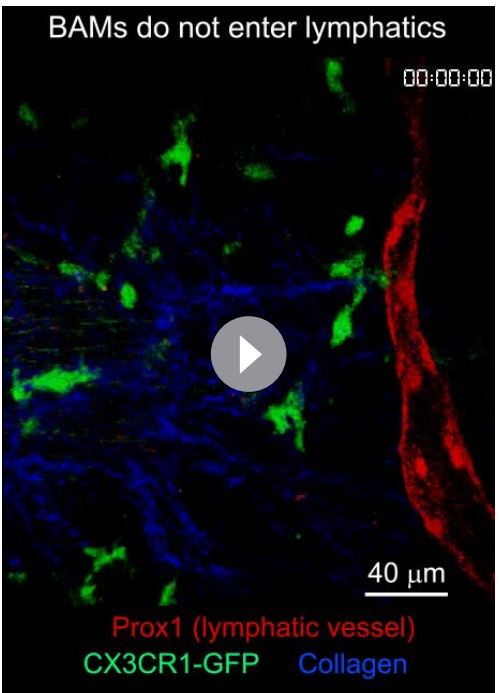

**Video 8.** Some DCs enter lymphatics. The video is a time-lapse image sequence showing CD11c-YFP[bright] DCs (green) migrating within a Prox1-tdTomato[+] lymphatic vessel (red). One DC can be seen that is entering the lymphatic vessel. Collagen (blue) was visualized by second harmonic generation. The image sequence was rendered from a 58 µm z-stack collected by two-photon microscopy. The mouse was administered intranasal LPS 2 hr before lung isolation, and then various precision-cut thick lung slices were imaged over a period of several hours prior to this recording. Elapsed time is shown as hh:mm:ss.
https://elifesciences.org/articles/63296/figures#video8

**Video 9.** BAMs do not enter lymphatics. The video is a time-lapse image sequence corresponding to Figure 7H, right panel. CX3CR1-GFP[bright] BAMs (green) can be observed proximal to a Prox1-tdTomato[+] lymphatic vessel (red), but do not enter the lymphatic. Collagen (blue) was visualized by second harmonic generation. The image sequence was rendered from a 32 µm z-stack collected by two-photon microscopy. The mouse was administered intranasal LPS 3.5 hr before lung isolation and then various precision-cut lung slices were imaged over the next few hours prior to this recording. Elapsed time is shown as hh:mm:ss.
https://elifesciences.org/articles/63296/figures#video9

where airflow changes directions, leading to the activation of adaptive immune responses at these locations. We found that these sites corresponded to the regions where inflammatory clusters form in allergic airway disease models. BALT has also been observed near airway bifurcations (*Bienenstock and McDermott, 2005*; *Randall, 2010*). Our observation that BAMs and DCs were enriched near airway branchpoints in the steady state, prior to immunization, suggests that airway branchpoints are ideal sites for continuous immunosurveillance of the bronchial airway lumen, enabling the rapid local activation of antigen-specific T cells and the generation of an inflammatory response. While most of the lung is filled with alveolar space to support gaseous exchange, these collagen-rich adventitial regions of connective tissue and interstitial space in the bronchovascular bundles represent a site where immune cells may be recruited. Recent work indicates that similar interstitial spaces may be found in other organs and that the adventitial 'cuff' regions may generally be important sites for tissue immunity (*Benias et al., 2018*; *Dahlgren and Molofsky, 2019*).

While both BAMs and DCs are strategically positioned in the lung for immunosurveillance, they differ substantially in the efficiency of antigen uptake in the steady state. Consistent with previous imaging studies of CD11c-YFP mice (*Thornton et al., 2012*; *Veres et al., 2011*), we observed minimal fluorescent antigen uptake by CD11c-YFP[+] DCs near the airways by microscopy. Our quantitative analysis of microscope images and flow cytometry revealed that CD11c-YFP[+] DCs were nevertheless taking up low amounts of soluble antigen. In contrast, BAMs took up higher amounts of antigen that

were readily visible by imaging. In a similar manner, IMs are superior at capturing soluble antigen from the intestinal lumen compared to DCs (*Mazzini et al., 2014*; *Schulz et al., 2009*). Although one group reported equivalent uptake of antigen by DCs and IMs in the lung (*Hoffmann et al., 2018*), this was likely due to contamination of their DC population, which was defined as CD11c+ MHC-II+ cells, by BAMs. We have shown here that BAMs express surface CD11c and MHC-II, yet BAMs exhibit distinct antigen uptake, morphology, and motility characteristics compared with cDCs. Considering that BAMs look more dendritic than lung cDCs and are MHC-IIhi, it is likely that the MHC-II+ dendriti-form cells identified in rodent airways in earlier studies (*Fear et al., 2011*; *Huh et al., 2003*; *von Garnier et al., 2007*) were actually BAMs rather than DCs. The stellate shape of BAMs is reminiscent of recent imaging studies of IMs in other tissues, such as in the kidney (*Stamatiades et al., 2016*), and may facilitate antigen sampling.

We observed that BAMs engaged in frequent, extensive interactions with DCs. The punctate local-ization of CD169 at the interacting surface between BAMs and DCs suggests this cell contact has biological significance, given precedent for a role for CD169 in antigen transfer from macrophages to DCs in the spleen and lymph nodes that contributed to cross-presentation (*van Dinther et al., 2018*). Phagocytosed antigen could potentially be passed from BAMs to DCs, similar to intestinal macro-phages which have been shown to pass antigen to DCs through gap junctions (*Mazzini et al., 2014*). Peptide-MHC complexes could also potentially exchange from BAMs to DCs through trogocytosis. BAMs could thus play an important role in immunosurveillance, both directly and through antigen transfer to DCs.

During inflammatory exposures, BAMs stay resident near the bronchial airways, whereas some DCs migrate to the draining lymph node. Consequently, BAMs may play an important role in the local activation of effector/memory T cells recruited to the lung. After initial sensitization, effector T cells were recruited to the airway interstitium where BAMs were abundant as early as 4–6 hr after allergen challenge and formed long interactions with BAMs, reminiscent of activating interactions previously observed between T cells and DCs in the lymph node (*Mempel et al., 2004*). These interactions may lead to the production of chemotactic factors that recruit additional cells to these sites forming inflammatory foci. Thus, the recruitment and activation of T cells at the bronchial airway interstitium, where BAMs display captured antigen, would provide a basis for the initiation of allergic inflammation during allergen re-exposure in sensitized individuals. In the intestine and adipose tissue, resident macrophages were also reported to activate effector T cells and shape local immunity (*Cho et al., 2014*; *Kamada et al., 2009*; *Schreiber et al., 2013*); BAMs may be the functional equivalent of these macrophages at the airways.

Here we reported the rapid and efficient uptake of soluble antigen across a range of sizes by BAMs, with the magnitude of antigen uptake inversely proportional to antigen size. We did not readily observe dendrite extension through the epithelial barrier by live microscopy, making this unlikely to account for the widespread antigen uptake by BAMs, and therefore suggesting that BAMs capture soluble antigens by alternative mechanism(s). Soluble antigens may instead be captured after diffusing across the airway epithelial barrier. This model may account for the reduced uptake of larger molec-ular weight antigens, which would be less effective at diffusing across the epithelium. In the intestine, it was reported that low molecular weight soluble antigens passed through goblet cells (*McDole et al., 2012*). Although few goblet cells are present in the murine lung in the steady state, whether soluble antigens could also pass through the more abundant club cells remains to be determined. The increase in goblet cells that occurs in allergic airway disease might also facilitate the passage of soluble antigens and thus further promote the inflammatory response. At sites of BALT formation, the follicle-associated epithelium containing microfold (M) cells has also been reported to facilitate the uptake of soluble antigens (*Bienenstock and Befus, 1984*). Alternatively, soluble antigen may pass through gaps in the airway epithelium, such as tri-junctional complexes located where three epithe-lial cells meet (*Patton, 1992*). The 200 nm bead particles that we tested would be too large to pass through the gaps in tri-junctional complexes that average 27 nm, consistent with our observation of minimal bead uptake around the airways by either BAMs or DCs. Interestingly, one study reported that very small 20 nm beads were captured by some MHC-II+ cells underneath the tracheal epithelium, although the majority of beads of a range of sizes were captured by AMs in the lung parenchyma (*Blank et al., 2013*). Indeed, we also observed that 200 nm beads were primarily captured within the alveolar spaces by AMs. Another study observed the uptake of 1000 nm beads by DCs located at the

alveoli, but not by DCs located near the conducting airways (*Thornton et al., 2012*). In that study, previously sensitized mice were rechallenged with antigen while beads were administered, creating an inflammatory state in which monocyte-derived cells and DCs (*Jenkins and Hume, 2014*; *Rodero et al., 2015*) likely were recruited to the alveolar regions and captured beads. In the steady state, DCs are largely restricted to the regions near bronchial airways, yet monocytes have also been reported to patrol the alveoli and capture beads (*Rodero et al., 2015*). Overall, our work reveals that at the bronchial interstitium, small soluble antigens are preferentially captured by BAMs.

The form and size of an antigen was recently reported to affect the type of T cell responses induced, as type 2 allergic responses were primarily directed to small soluble antigens whereas regulatory T cell responses were primarily directed to particulate antigens (*Bacher et al., 2016*), though the mechanism was not explored. Our finding that soluble and particulate antigens are preferentially captured in different regions of the lung leads us to propose that the anatomic location of antigen uptake ultimately influences the final T cell response. The tissue signals inherent in different anatomic environments may prime divergent immune responses. In the alveoli, the abundant AMs have been proposed in several studies to dampen allergic responses (*Bang et al., 2011*; *Mathias et al., 2013*; *Tang et al., 2001*; *Zasłona et al., 2014*). Macrophages that were earlier found in the alveolar interstitium were also reported to produce IL-10 and reduce inflammation in allergic airway disease models (*Bedoret et al., 2009*; *Kawano et al., 2016*). The tissue environment at the alveoli thus appears to favor reduced inflammation in the delicate alveolar tissue, in the absence of overt infection in this region. Instead, the airways may be more permissive to the induction of inflammation as indicated by the concentration of immune responses at airway adjacent regions. Future work on the functional capacities and/or cytokine profiles of BAMs would further inform our understanding of the inflammatory milieu at the airway microenvironment.

We propose that BAMs represent a subset of IMs specialized for sampling the bronchial airway lumen. This work extends the findings of *Gibbings et al., 2017* and *Rodero et al., 2015* who first observed CX3CR1-GFP+ IMs localized near the bronchial airways. IM populations near the alveoli have also been reported in microscopy studies (*Bedoret et al., 2009*; *Schyns et al., 2019*; *Tan and Krasnow, 2016*). Multiple subsets of IMs have also been described based on different combinations of surface markers by flow cytometric analysis, as well as distinct transcriptional profiles (*Chakarov et al., 2019*; *Gibbings et al., 2017*; *Sabatel et al., 2017*; *Schyns et al., 2019*). There is a lack of consensus on the identification and localization of the IM subsets, and it seems likely that diverse pathogen exposures in different mouse facilities may affect baseline inflammation and the relative abundance of different subsets. This may explain, for example, why we observed few CD169[bright] macrophages near the airways and nerves, whereas another group found these were abundant (*Ural et al., 2020*). Of note, most of our mice for this study were bred in a 'super-barrier' mouse facility that excludes additional pathogens compared to standard SPF facilities. Here we have focused specifically on the subpopulation of IMs localized near the bronchial airway epithelium that we found readily captures soluble inhaled antigens in naïve mice. We propose that the field consider referring to these macrophages as BAMs based on their anatomic localization near the bronchial airways, much as it has been useful to classify AMs based on their anatomic location. We found that BAMs expressed high amounts of MHC-II; were efficient at antigen capture, processing and presentation; and interacted with effector T cells in vivo; suggesting that BAMs serve as APCs near the bronchial airways. Our finding that BAMs expressed high amounts of surface MHC-II differs from a recent study by *Schyns et al., 2019*, which reported MHC-II+ CX3CR1-GFP+ IMs were primarily localized near the alveoli. This difference is likely due to technical differences in our imaging approaches. We imaged native GFP fluorescence in unfixed tissue allowing us to preferentially visualize CX3CR1-GFP[bright] BAMs, whereas Schyns et al. used anti-GFP antibody amplification of CX3CR1-GFP signal in fixed tissue, which would have likely resulted in the visualization of a much larger population of cells with varying expression of CX3CR1-GFP, including monocytes and DCs. We also imaged thick tissue vibratome sections by two-photon microscopy whereas Schyns et al. imaged thin cryosections by confocal microscopy. We found the analysis of thick sections to be particularly useful to try to discern the cellular distribution of BAMs in unimmunized mice, whereas in thin sections the cells were too sparse.

Emerging data from microarray and RNAseq analyses have highlighted the distinct gene expression patterns of IMs, AMs and DCs in the lung (*Chakarov et al., 2019*; *Gautier et al., 2012*; *Gibbings et al., 2017*; *Schlitzer et al., 2013*; *Schyns et al., 2019*), which we have further elaborated here

by our scRNAseq analysis of CD11c+ cells. Functional studies in other tissues such as the intestine have revealed distinct contributions of macrophages and DCs to immunity (*Chang et al., 2013*; *Kamada et al., 2009*; *Mazzini et al., 2014*; *Schreiber et al., 2013*; *Schulz et al., 2009*; *Shaw et al., 2012*). Here, we have shown that BAMs, as a subset of IMs, are distinct from lung cDCs in morphology, antigen uptake capacity and the ability to migrate to the draining lymph node. It is important to note that under inflammatory conditions, including in mouse models of allergic airway disease, monocytes rapidly enter lung tissue and differentiate into monocyte-derived cells that express surface markers that bear some resemblance to resident DCs and macrophages (*Guilliams et al., 2013*; *Jakubzick et al., 2008*; *Plantinga et al., 2013*). Although the MAR-1 antibody to FcεRI was thought to be a marker of these monocyte-derived cells, we recently established that this antibody also binds to CD64 and FcγRIV (*Tang et al., 2019*), and thus would also detect resident CD64+ macrophages. Recent studies of respiratory viral infection have also revealed the recruitment of a subpopulation of cDCs to the lung that stain with MAR-1 and express CD64 (*Bosteels et al., 2020*), further confounding the use of these markers to distinguish macrophages and cDCs in the context of inflammation. One microscopy study of the lungs of mice administered HDM reported an increase in the abundance of CD64+ "DCs" identified by CD11c and MHC-II expression (*Hoffmann et al., 2018*), which likely consisted of a heterogeneous mixture of cDCs, resident BAMs and monocyte-derived cells. Another study used an extensive analysis of the ratio of expression of CX3CR1-GFP and CD11c-mCherry reporters to determine the relative differentiation state of infiltrating monocytes (*Sen et al., 2016*). This paper showed an increase in the abundance of cells with high expression of both CX3CR1-GFP and CD11c-mCherry reporters in allergic airway inflammation models. These cells were proposed to represent differentiated monocyte-derived cells, and we speculate that the rare cells we observed that were bright for CX3CR1-GFP and CD11c-YFP reporters in naïve mice may also represent cells that had recently differentiated from monocytes. One challenge, however, in relating this study to our findings presented here is that the high expression of CX3CR1-GFP in resident BAMs versus the lower expression in monocytes was not considered in the analysis of the ratio of CX3CR1-GFP to CD11c-mCherry expression, which formed the basis of the monocyte differentiation model (*Sen et al., 2016*). Similar to BAMs and DCs, monocyte-derived cells have been reported to activate T cells and drive allergic inflammation (*Plantinga et al., 2013*). Conversely, monocyte-derived cells and IMs have also been reported to promote tolerance and dampen inflammation in the lung (*Bedoret et al., 2009*; *Kawano et al., 2016*; *Sabatel et al., 2017*). Although all these APCs may exhibit some overlap in their ability to activate T cells, it seems likely that these different cell types may skew T cell responses at the airways in distinct ways. Further studies with selective tools for manipulation will be needed to elucidate the specific contributions of BAMs, DCs, and monocyte-derived cells to lung immunity. While we have focused on the potential role of BAMs as APCs in immune surveillance of the bronchial airways, our findings do not exclude other potential functions of these cells. The association of some BAMs with nerves at bronchovascular bundles is likely of significance to tissue function and/or immunity. BAMs might also be involved in the clearance of epithelial cells or in the production of molecules involved in tissue homeostasis. It will be interesting for future studies to resolve the potential distinctions and functions of the various subsets of IMs recently defined in the lung.

In summary, our study demonstrates that the bronchial airways are a major site of soluble antigen capture where BAMs serve as resident sentinel immune cells. Compared to DCs, BAMs exhibited distinct morphology and enhanced soluble antigen uptake from the airways. BAMs also closely interacted with cDCs, suggesting the potential for antigen transfer between these cell types. In addition, DCs migrated to the draining lymph nodes whereas BAMs remained resident near the airways. This disparity in migratory behavior suggests a division of labor where migratory cDCs may prime naïve T cells in the lymph nodes while BAMs mediate local airway immunity. Our data demonstrate that BAMs are able to activate effector Th2 cells and interact with CD4 T cells recruited to the airways in mouse models of allergic airway disease. We propose that BAMs may play a similar role in the local activation of T cells in context of infection. Overall, BAMs are strategically positioned as sentinel APCs underlying the bronchial airway epithelium. BAMs were enriched at collagen-rich regions near some airway branchpoints, which may ultimately develop inflammatory foci when an adaptive immune response is initiated.

## Methods

### Mice and chimeras

C57BL/6 J (B6/J; 000664), μMT (002288; B6.129S2-*Ighm*[tm1Cgn]/J), CX3CR1-GFP (005582; B6.129P-*Cx-3cr1*[tm1Litt]/J), CD4-Cre (022071; B6.Cg-Tg(Cd4-cre)[1Cwi/Bflu]/J), Shh-Cre (005622; B6.Cg-*Shh*[tm1(EGFP/cre)Cjt]/J), Ai14 (007914; B6.Cg-Gt(*ROSA*)26Sor[tm14(CAG-tdTomato)Hze]/J), E-cadherin-mCFP (016933; B6.129P2(Cg)-*Cdh1*[tm1Cle]/J), MacBlue (026051; STOCK Tg(Csf1r*-GAL4/VP16,UAS-ECFP)[1Hume]/J), ROSA26-mTmG (007676; B6.129(Cg)-*Gt(ROSA)26Sor*[tm4(ACTB-tdTomato,-EGFP)Luo]/J), Prox1-tdTomato (022766; C57BL/6-Tg(Prox1-tdTomato)12[Nrud]/J), Thy1.1 (000406; B6.PL-*Thy1*[a]/CyJ) and Zbtb46-GFP (018534; 129S-*Zbtb46*[tm1.1Kmm]/J) mice were originally from The Jackson Laboratories. All mice were maintained on B6 background except Zbtb46-GFP, which was obtained on a 129 background and crossed one generation to B6/J. MacGreen (CSF1R-GFP, Tg(Csf1r-EGFP)[1Hume]) (*Sasmono et al., 2003*), CD11c-YFP (Tg(Itgax-Venus)1[Mnz]) (*Lindquist et al., 2004*), β-actin-CFP (Tg(CAG-ECFP)CK6[Nagy]) (*Hadjantonakis et al., 2002*), and OT-II (Y-chromosome insertion; Tg(TcraTcrb)426-6[Cbn]) (*Barnden et al., 1998*) mice were maintained on a B6 background. B6-CD45.1 congenic mice, which were used as wild-type mice in some experiments, were bred from the Boy/J line (002014; B6.SJL-*Ptprc*[a]*Pepc*[b]/BoyJ) from The Jackson Laboratories or purchased from Charles River (National Cancer Institute Model 01B96; B6-Ly5.2/Cr, later renamed to B6-Ly5.1/Cr). All mice were maintained in specific-pathogen-free facilities and protocols were approved by the Institutional Animal Care and Use Committee of the University of California San Francisco. MacGreen mice were crossed to μMT mice to generate a faithful reporter of myeloid cells because some B cells were also GFP[+] (unpublished observations). Fluorescent OT-II T cells were obtained by crossing OT-II mice to β-actin-CFP mice or CD4-Cre X Ai14 mice.

### Antigen administration

Intranasal (i.n.) droplets were administered to mice that had been lightly anesthetized with isoflurane using a controlled vaporizer. To induce allergic airway inflammation with a classical OVA model (*Daubeuf and Frossard, 2013*) for *Figure 1A*, mice were first sensitized by intraperitoneal (i.p.) injections with 100 μg Endograde OVA (Hyglos) mixed with 100 μl alum (Alhydrogel, Grade A 1.3%, Accurate Chemical and Scientific) once a week for a total of three times. One week after the last sensitization, mice were challenged i.n. with 50 μg Endograde OVA in 50 μl PBS daily over three days. Lungs (see details below) were assessed two days after the last challenge. For *Figure 1C*, 10[4] OT-II cells were adoptively transferred i.v. and then mice were primed i.p. with 100 μg OVA grade VII (Sigma-Aldrich, this lot was found to have minimal endotoxin activity) in 200 μl alum on d0 and d9, and then challenged i.n. with 100 μg OVA in 50 μl PBS on d16, 17, and 18, and then imaged on d20. In repeats of this particular experiment, the mice were primed once rather than twice with OVA/alum, as this led to greater recovery of OT-II cells and a similar inflammatory response. For experiments represented in *Figure 1D*, mice received 10–100 μg of HDM extract (Greer) reconstituted in PBS i.n. once a week for a total of four times (*Köhl et al., 2006*). Lung slices were imaged 2 days after the last exposure. For live imaging, an abbreviated i.n. model of OVA allergic airway inflammation (*Eisenbarth et al., 2002*) was used to study early events after allergen challenge. Mice were sensitized i.n. with 100 μg Endograde OVA mixed with low-dose (100 ng) LPS (ultrapure LPS, O55:B5, List Biologicals) in 50 μl of PBS and OT-II cells were adoptively transferred i.v. on the same day. Nine to 10 days later, mice were then challenged i.n. with 50 μg OVA in 50 μl of PBS, followed by excision of lungs 4–6 hr after challenge.

For microscopy, fluorescent OVA conjugated to Texas Red or Alexa 647 (Life Technologies) was administered in 20–50 μl PBS i.n. Smaller volumes were found to somewhat favor labeling at the larger airways rather than at the alveoli, but also led to a more heterogeneous distribution of antigen throughout the lung. To produce HDM conjugated to Texas Red or OVA conjugated to TAMRA, 100 μg of HDM or OVA were reconstituted in PBS with 0.1 M sodium bicarbonate at 1 mg/ml and mixed with 18 μg Texas Red-succinimidyl ester or 36 μg 5,(6)-TAMRA-succinimidyl ester (Life Technologies), respectively. Excess free Texas Red or TAMRA were removed with Bio-spin 6 kDa size exclusion columns (Bio-Rad Laboratories) equilibrated with PBS according to manufacturer's instructions. We noted that Texas Red-labeled antigens bound nonspecifically to dead epithelial cells at the cut site of the vibratome sections, but this was not observed with TAMRA- or Alexa 647-labeled antigens. However, we observed similar capture of all types of fluorophore-labeled antigens by BAMs, confirming this result was not fluorophore-dependent. TRITC-Dextran and FITC-Dextran were from

Life Technologies (10 kDa) or Sigma-Aldrich (all other molecular weights). R-Phycoerythrin (PE) (Life Technologies) was dialyzed for 48 hr at 4 °C into PBS using 10 kDa MWCO Slide-a-lyzer cassettes (Thermo Fisher) before i.n. administration. 0.2 µm Sky Blue fluorescent beads (Spherotech) were administered i.n. at 0.025–0.05% w/v in 30 µl PBS. To induce CCR7 upregulation and track lymphatic migration (*Figure 8*), mice received either 1 µg ultrapure LPS or 100 µg HDM in 30–50 µl of PBS i.n., or were primed and challenged with the abbreviated low-dose LPS i.n. OVA model described above, and lungs were analyzed 4–6 hr later.

To generate OVA-Eα, an Eα peptide (*Murphy et al., 1992*) synthesized with an added terminal cysteine residue (New England Peptide) was dissolved in ultrapure water with 5 mM EDTA and mixed with maleimide-activated OVA (Thermo Fisher Scientific) at room temperature for 2 hr. Excess Eα peptide was removed with Bio-spin 30 kDa size exclusion columns (Bio-Rad Laboratories) equilibrated with PBS according to the manufacturer's instructions. Twenty µg OVA-Eα was administered i.n. together with 2.5 µg of OVA-Alexa 647 (Life Technologies) in 30–50 µl PBS and lungs were analyzed 17–18 hr later. For imaging of T cell activation, $5\times10^5$ fluorescent OT-II T cells (see Mice and Chimeras above) were adoptively transferred i.v. into the retro-orbital plexus of anesthetized CX3CR1-GFP X CD11c-YFP recipient mice, which were then sensitized and challenged with the abbreviated low-dose LPS i.n. OVA model described above, and lungs were excised for imaging 4–6 hr after challenge.

## Cryosection and immunofluorescence microscopy

Mouse lungs were inflated with 1 mL of Tissue-Tek optimum cutting temperature compound (OCT, Sakura), then placed in a cryomold with OCT and 'snap-frozen' in dry ice and ethanol. Frozen tissue blocks were stored at –80 °C. Acetone-fixed cryosections were prepared as described (*Allen et al., 2004*). Slides were stained as described (*Yang et al., 2018*) in PBS or TBS containing 0.1% BSA (Roche), 1% normal mouse serum (Jackson Immunoresearch) and a diluted mixture of purified Siglec-F PE (E50-2440, BD) and SMA-FITC (1A4, Abcam) for 2 hr. Slides were washed for 3 min in PBS or TBS and mounted in Fluoromount G with DAPI (SouthernBiotech). Slides were imaged on the Axio Scan. Z1 slide scanner equipped with an HXP120 fluorescence microscope illuminator as well as 38 HE and 20 HE filter sets for FITC and PE visualization (Carl Zeiss Microscopy). A Chroma filter set (GT370/60 x, T425lpxr, ET460/50 m) was used for DAPI visualization. Images were collected and analyzed with Zen Blue (Carl Zeiss Microscopy).

## Two-photon microscopy

In preparation for 2-photon microscopy, mouse lungs were inflated with 1 mL of 2% low melt agarose, and 250–600 µm thick sections were precision cut using a vibratome (Leica VT1200 or Oxford Laboratories Model G) as described (*Sullivan et al., 2011*) and then placed in Leibovitz's L-15 medium at room temperature until imaging. Airways could be identified by transmitted light based on the presence of a columnar epithelial lining and active ciliary beating. Imaging was performed within several hours with the exception of slices stained with antibodies. For antibody staining, lung slices were incubated at room temperature in PBS with TruStain FcX (Biolegend) diluted to 5 µg/mL, mouse gamma globulin (Jackson ImmunoResearch) diluted to 100–400 µg/mL, and in some cases rat gamma globulin (Jackson ImmunoResearch) diluted to 100–200 µg/mL for 2 hr and then stained with the antibodies (see *Supplementary file 1*) overnight at 4 °C. Lung slices were then washed in excess PBS before imaging. For time-lapse microscopy, lung slices were incubated as described (*Sullivan et al., 2011*). Briefly, lungs slices were mounted in a heated chamber (JG-23W/HP, Warner Instruments) with warm RPMI (Gibco) supplemented with 1 X Penicillin, Streptomycin, and L-Glutamine (Life Technologies) flowing over it at a rate of 1.5 mL/min. Media temperature in the imaging chamber was maintained at 35–37°C. Carbogen (5% $CO_2$/95% $O_2$) was bubbled into RPMI to maintain physiologic pH.

Lung slices were imaged on an LSM 7 MP INDIMO two-photon laser scanning microscope (Carl Zeiss Microscopy) equipped with two Chameleon Ultra II lasers (Coherent), a Compact Optical Parametric Oscillator (OPO, Coherent) and a W Plan-Apochromat 20 x/1.0 objective. Images were collected on non-descanned detectors consisting of four gallium arsenide phosphide (GaAsP) detectors and one far-red sensitive detector. Transmitted illumination was detected on a T-PMT. Emission filters used to record fluorescence signals are listed in *Supplementary file 2*. Wavelengths used to excite particular fluorophores are listed in *Supplementary file 3*. Sequential laser excitation at different wavelengths was used to enhance the spectral separation of fluorophores, similar to our previous

approach (*Sullivan et al., 2011*) with CFP and YFP. Samples were imaged with multiple tracks, with rapid line or frame switching of laser emissions by acousto-optical modulators in order to separate the excitation and detection of different fluorophores and for second harmonic generation. For example, in order to improve the separation of CFP and GFP from YFP: in the first track, one laser was tuned to 850–870 nm to excite CFP and GFP with minimal excitation of YFP; and then in the second track, the other laser was tuned to 1020–1040 nm to strongly excite YFP with minimal excitation of CFP and GFP. In some cases, a third track was added with the OPO tuned to 1100–1200 nm to strongly excite PE and/or Alexa 647 with minimal excitation of CFP/GFP/YFP. Second harmonic generation results in an emission at half the incident laser wavelength and the signal was collected in various filters (*Supplementary file 2*); the separation of the second harmonic from fluorophore emissions was assisted by tuning lasers to particular wavelengths and the use of multiple tracks. For experiments in which enhanced spectral separation was not needed, or for fast imaging for time-lapse recordings, fluorophores were excited simultaneously with one or more laser lines. Z-stacks were collected every 30 s for 20–45 min. *xy* planes were collected at variable zoom and pixel sizes, with a typical resolution of 512x512 pixels (some high resolution images were collected at 1,024x1,024 pixels). Bidirectional scanning was typically used for faster image acquisition. The spacing between planes in the *z* dimension was typically set to three to five times the *xy* pixel size for a given zoom setting. Images were collected with Zen Black (Carl Zeiss Microscopy) and images were analyzed with Zen Black, Velocity (PerkinElmer) and Imaris software (Bitplane). Videos were prepared with Velocity and then annotated in Adobe After Effects CC.

## Flow cytometry

To collect bronchoalveolar lavages (BAL), mice were euthanized with an overdose of isoflurane and lungs were deflated and exposed by thoracotomy. A 20Gx1" SurFlash catheter (Terumo) was inserted into the trachea and secured with a suture. Lungs were flushed twice with a syringe with 1 mL of DPBS (Gibco) and then the lavage was added to 200 µl of PBS with 50% FBS and 0.1 M EDTA (Gibco) and kept on ice. BALs were centrifuged at 350 g for 7 min at 4 °C and resuspended in PBS with 2% FBS, 1 mM EDTA (Gibco) and 0.1% sodium azide (Gibco) (FACS buffer) for antibody staining. Lungs were excised into 1 mL DMEM media supplemented with HEPES (Gibco), Penicillin, Streptomycin and L-Glutamine (Life Technologies). For lung digestion, 25 µg of DNase I (Sigma) and 0.28 Wunitz Units of Liberase TM (Roche) were added to each mL of digest. Lungs were snipped into small pieces with scissors in a 1.7 mL eppendorf tube and incubated on a 37 °C shaker for 30–45 min at 700 rpm on a ThermoMixer F1.5 (Eppendorf). The digestion reaction was quenched by adding FBS and EDTA to a final concentration of 5% FBS and 10 mM EDTA. Digested lungs were then gently dissociated and passed through a 70 µm strainer to yield single cell suspensions that were then centrifuged and resuspended in DMEM supplemented with 5% FBS, HEPES, Penicillin, Streptomycin and L-Glutamine. For FACS staining, $3x10^6$ lung cells were added to a 96-well plate and washed with 150 µl of ACK Lysing Buffer (Quality Biological) before antibodies were added for staining. Flow cytometric staining of BAL and lung cells was done as described (*Allen et al., 2004*; *Yang et al., 2012*), except CCR7 was stained for 15 min at 37 °C. Antibodies used for staining are listed in *Supplementary file 1*. Flow cytometric data was acquired on a BD LSRFortessa with 405 nm, 488 nm, and 640 nm lasers, using BD FACS DiVa software and analyzed with FlowJo software (TreeStar).

## scRNAseq analysis

Lungs were excised from four naïve mice (two males and two females) and digested and dissociated as above. MacBlue mice were used to allow for the analysis of CFP expression in future studies. Single cell suspensions of two lung samples from mice of the same sex were pooled, then a 2 X volume of ACK Lysing Buffer (Quality Biological) was added to lyse red blood cells, then cells were centrifuged at 350 g for 7 min at 4 °C, and then cells were resuspended in PBS with 0.5% BSA (Roche) and 2 mM EDTA (Gibco) (MACS buffer). Cells were depleted of B cells and partially depleted of AMs and cells that had lost membrane integrity by first incubating with anti-IgD Biotin (Southern Biotech) and anti-Siglec-F Biotin, as well as 0.25 µM of NHS-Biotin (Life Technologies, freshly reconstituted in PBS), for 30 min on ice. Cells were washed in MACS buffer and incubated with 50 µl of MyOne Streptavidin T1 Dynabeads (Thermo Fisher) for 30 min, during which tubes were inverted and rotated at 4 °C. Tubes were placed in a Dynamag-2 magnet (Thermo Fisher) for 2 min for bead isolation and the negative

fraction containing unlabeled cells was collected and diluted with MACS buffer to 5 ml. To further enrich for macrophages and DCs and to remove stromal cells and dead cells, 3 ml of 14% OptiprepTM (Axis-Shield) was layered below the cell suspension, and cells were centrifuged at 600 g for 20 min with no brake at 8 °C. Cells at the interface were collected and washed in MACS buffer. Finally, CD11c$^+$ cells were enriched by incubating with CD11c Mojosort Nanobeads (Biolegend) for 15 min at 4 °C. Bead-labeled cells were purified with the Mojosort magnet (Biolegend) according to the manufacturer's instructions. Bead-labeled cells were washed in PBS with 0.5% BSA (Roche), washed, and submitted to the UCSF Institute for Human Genetics for single cell RNA sequencing using 10 x Genomics Chromium technology. Data was processed in Cell Ranger, v5.0.1 (10x Genomics) through count and aggregate pipelines according to manufacturer's instructions and visualized using Loupe Cell Browser 5.0 (10x Genomics).

## APC sorting

Lungs were excised from CD45.1 congenic mice into DMEM media supplemented with HEPES, Penicillin, Streptomycin and L-Glutamine and digested and dissociated as above. Single cell suspensions of lung digests were combined and then a 2 X volume of ACK Lysing Buffer (Quality Biological) was added and cells were centrifuged at 350 g for 7 min at 4 °C and resuspended in PBS with 0.5% BSA (Roche) and 1% FBS (Sorting Buffer). Cells were incubated with CD11c BV650 antibody together with CD11c Mojosort Nanobeads (Biolegend), after which bead-labeled cells were purified with the Mojosort magnet (Biolegend) according to the manufacturer's instructions. Cells were then washed and stained with sorting antibodies (Viability dye eFluor780, CD45 Alexa 700, Ly6G PE-Cy7, Siglec-F BV421, CD19 PE-Dazzle 594, CD11b BV785, MHC-II I-A$^b$ FITC, CD64 PE, CD24 APC) at $4 \times 10^7$ cells/ml. Cells were washed and diluted with Sorting Buffer, centrifuged through a 1 mL layer of 100% FBS, resuspended in Sorting Buffer and passed through a 40 μm cell strainer. Cells were sorted on the FACSAria III (BD) into Advanced RPMI (Gibco) with 20% FBS.

## T cell culture

Inguinal, brachial, axillary and cervical lymph nodes were excised from OT-II mice with a Thy1.1 congenic marker. The tissues were then gently dissociated and passed through 40 μm strainers to yield single cell suspensions that were centrifuged at 350 g for 7 min at 4 °C and resuspended in DMEM supplemented with 5% FBS, HEPES, Penicillin, Streptomycin and L-Glutamine. To purify CD4 T cells by negative selection, cells were resuspended at $1–2 \times 10^8$ cells/ml and incubated with biotinylated antibodies against B220, CD11b, CD11c, CD8, NK1.1, and MHC-II I-A$^b$ for 30 min on ice. Cells were washed to remove excess antibodies and then incubated with 250 μl MyOne Streptavidin T1 Dynabeads (Thermo Fisher) per $10^8$ cells for 30 min, during which tubes were inverted and rotated at 4 °C. Tubes were placed in a Dynamag-2 magnet (Thermo Fisher) for 2 min for bead isolation and the negative fraction containing unlabeled cells was collected and washed. For Cell Trace Violet labeling, cells were centrifuged and resuspended in PBS with 2% FBS at $2–20 \times 10^6$ cells/ml. A 10 mM stock of Cell Trace Violet (Thermo Fisher) in DMSO was diluted to 10 μM in PBS and added at 1:1 volumetric ratio to cells while vortexing. Cells were incubated with Cell Trace Violet for 20 min at 37 °C, then 100 μl of 100% FBS was layered at the bottom of the tube and cells were then centrifuged and resuspended in Advanced RPMI (Gibco) culture media. 50,000 labelled naïve T cells were co-cultured with 10,000 sorted APCs for 2.5 days in the presence or absence of 20 μg/mL OVA$_{323-339}$ peptide (Invivogen) before analysis.

For Th2 polarization, CD4 T cells were purified by negative selection with magnetic beads as described above and cultured in Advanced RPMI (Gibco) supplemented with 10% FBS, HEPES, Penicillin, Streptomycin, L-glutamine, 1 mM sodium pyruvate and 50 mM 2-mercaptoethanol (culture media). Flat-bottom 96-well plates were coated with 2.5 μg/mL anti-mouse CD3e (eBioscience, functional grade purified, eBio500A2) in PBS for 3 hr at 37 °C and washed twice with PBS before use. T cells were added to anti-CD3 coated wells at $10^6$ cells/mL in the presence of 1 μg/mL anti-CD28 (Biolegend, LEAF purified, 37.51), 100 U/ml IL-2 (gift from K. Mark Ansel's lab), 50 ng/mL IL-4 (Peprotech) and 10 μg/ml anti-IFNγ (Biolegend, LEAF purified) for 3 days. T cells were then transferred into uncoated wells and replenished with IL-2, IL-4 and anti-IFNγ for another 3 days before co-culturing with sorted APCs. 50,000 T cells were co-cultured with 10,000 sorted APCs in 96-well V-bottom plates (Cellstar) in the presence or absence of 20 μg/mL OVA$_{323-339}$ peptide (Invivogen). To assess cytokine

production, 3 µg/mL Brefeldin A (eBioscience) was added to cultures 1–2 hr after the start of culture, for 14–18 hr before flow cytometric analysis. T cells were analyzed by flow cytometry as described above. For intracellular cytokine staining, cells were fixed and permeabilized with Cytofix/cytoperm solution (BD) for 20 min on ice and then washed with Perm/Wash solution (BD) according to the manufacturer's instructions. Cells were stained with anti-cytokine antibodies (*Supplementary file 1*) in Perm/Wash solution for 20 min on ice. Cells were washed twice with Perm/Wash solution and resuspended in FACS buffer, acquired on LSR Fortessa (BD) using FACS DiVA software (BD) and analyzed with FlowJo software (Treestar).

## Statistical analysis

Statistical analyses and graphical representations were done using GraphPad Prism. Data were expected to have a log-normal distribution and all data were log transformed for parametric statistical tests. The appropriate tests were chosen based on experimental design after consulting the GraphPad Statistics Guide. All tests were two-tailed. Multiple biological replicates and imaging fields were analyzed to draw conclusions. The number of samples analyzed was based on the expected magnitude of differences, which was estimated from past similar experiments or pilot studies. Results were further validated by repeating experiments and/or by performing related experiments. Detailed information on samples and experiments is provided in each figure legend. Full statistical results with exact p values for all figures are provided in *Supplementary file 4*.

## Acknowledgements

We thank D Sheppard for providing CD11c-YFP mice; Z Werb for providing β-actin-CFP and MacGreen mice; K M Ansel, H Chapman, J Cyster, R Locksley, J-S Shin, D Sheppard, and X Huang for helpful advice; J Gordon and the UCSF Laboratory for Cell Analysis (supported by National Cancer Institute Cancer Center Support Grant P30CA082103) for cell sorting; and M Shenoy for contributions to some early experiments. Research reported in this publication was supported by the National Heart, Lung, and Blood Institute (DP2HL117752) and the National Institute of Allergy and Infectious Diseases (R21AI130495) of the National Institutes of Health, the UCSF Cardiovascular Research Institute, the UCSF Sandler Asthma Basic Research Center, and the UCSF Program for Breakthrough Biomedical Research, which is partially funded by the Sandler Foundation. X.T. was funded by the Agency for Science, Technology and Research (A*STAR) National Science Scholarship (PhD). L.S.M.K. was supported by training grants from the National Institute of Allergy and Infectious Diseases (T32AI007334-31) and the National Heart, Lung, and Blood Institute (T32HL007731-28). The content is solely the responsibility of the authors and does not necessarily represent the official views of the funding agencies.

## Additional information

### Funding

| Funder | Grant reference number | Author |
| --- | --- | --- |
| National Heart, Lung, and Blood Institute | DP2HL117752 | Christopher D C Allen |
| National Institute of Allergy and Infectious Diseases | R21AI130495 | Christopher D C Allen |
| UCSF Cardiovascular Research Institute | | Christopher D C Allen |
| UCSF Sandler Asthma Basic Research Center | | Christopher D C Allen |
| Agency for Science, Technology and Research | | Xin-Zi Tang |
| National Institute of Allergy and Infectious Diseases | T32AI007334-31 | Lieselotte S M Kreuk |

| Funder | Grant reference number | Author |
|---|---|---|
| UCSF Program for Breakthrough Biomedical Research | | Christopher D C Allen Ross J Metzger |
| National Heart, Lung, and Blood Institute | T32HL007731-28 | Lieselotte S M Kreuk |

The funders had no role in study design, data collection and interpretation, or the decision to submit the work for publication.

### Author contributions

Xin-Zi Tang, Conceptualization, Formal analysis, Validation, Investigation, Visualization, Methodology, Writing – original draft, Writing – review and editing; Lieselotte S M Kreuk, Validation, Investigation, Writing – review and editing; Cynthia Cho, Investigation; Ross J Metzger, Investigation, Methodology, Writing – review and editing, provided critical insights into lung anatomy and methodology; assisted with imaging of airway bifurcations; Christopher D C Allen, Conceptualization, Formal analysis, Supervision, Funding acquisition, Validation, Investigation, Visualization, Methodology, Writing – original draft, Writing – review and editing

### Author ORCIDs

Xin-Zi Tang ⬤ http://orcid.org/0000-0001-6064-8624
Lieselotte S M Kreuk ⬤ http://orcid.org/0000-0003-4906-039X
Christopher D C Allen ⬤ http://orcid.org/0000-0002-1879-9047

### Ethics

The care, maintenance, and experimental manipulation of mice followed guidelines established by the Institutional Animal Care and Use Committee of the University of California, San Francisco under approved protocols AN079036, AN089524, AN111286, AN175836, and AN191685.

### Decision letter and Author response

Decision letter https://doi.org/10.7554/eLife.63296.sa1
Author response https://doi.org/10.7554/eLife.63296.sa2

## Additional files

### Supplementary files

• Supplementary file 1. Table of antibodies and reagents used for flow cytometry and microscopy.

• Supplementary file 2. Table of emission filters for 2-photon microscopy.

• Supplementary file 3. Table of excitation wavelengths for 2-photon microscopy.

• Supplementary file 4. Table of exact p values.

• Transparent reporting form

### Data availability

Relevant data are included in the manuscript figures and examples of 3D visualizations and time-lapse imaging are provided as videos. The RNAseq data have been deposited at the NCBI Gene Expression Omnibus (GEO) and are accessible through GEO Series accession number GSE214177 (https://www.ncbi.nlm.nih.gov/geo/query/acc.cgi?acc=GSE214177).

The following dataset was generated:

| Author(s) | Year | Dataset title | Dataset URL | Database and Identifier |
|---|---|---|---|---|
| Tang XE, Allen CDC | 2022 | Single cell gene expression profile of CD11c-enriched fraction of dissociated mouse lung immune cells | https://www.ncbi.nlm.nih.gov/geo/query/acc.cgi?acc=GSE214177 | NCBI Gene Expression Omnibus, GSE214177 |

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
