## [Editor Report]

The study here presented interesting evidence on the nature of a cell population responsible for antigen uptake and presentation in the airway and how this cell population could stimulate a Th2 response locally. Authors performed multi photon imaging of live tissue or 3D reconstruction of thick lung sections from various transgenic mouse models. The manuscript provides an elegant set of data that consolidate the study focusing on BAMs, their strategic positioning near the airways, the characterization of antigen capture and presentation, as well as migratory properties, as compared to DCs.

---

## [Decision Letter]

**Decision letter after peer review:**

[Editors’ note: the authors submitted for reconsideration following the decision after peer review. What follows is the decision letter after the first round of review.]

Thank you for submitting your work entitled "Bronchus-associated macrophages efficiently capture and present soluble inhaled antigens for local Th2 cell activation" for consideration by *eLife*. Your article has been reviewed by 3 peer reviewers, and the evaluation has been overseen by a Reviewing Editor and a Senior Editor. The reviewers have opted to remain anonymous.

Our decision has been reached after consultation between the reviewers. Based on these discussions and the individual reviews below, we regret to inform you that your work will not be considered further for publication in *eLife*.

The study here presented interesting evidence on the nature of a cell population responsible for antigen uptake and presentation in the airway and how this cell population could stimulate a Th2 response locally. Authors used various transgenic mice models to selectively deplete DC or macrophages and performed multi photon imaging of live tissue or 3D reconstruction of thick lung sections from these mouse models. However, all the reviewers concluded that the results presented here were too preliminary and lacking substantial argument on the novelty of the characterised population (overlap with lung IM? no RNA-seq characterisation) as well as the quantitative aspect of the data reported here by imaging and finally the claim on the local Th2 cell activation performed by this population. We all discussed these matters and estimated that too many additional experiments would be necessary to further address these very important questions. Please find below the comments of the reviewers that we hope that you will find helpful.

*Reviewer #1:*

The authors address the question of antigen uptake focusing on the bronchus, with the rational that it represents the main site of antigen deposit. They challenge the idea that contrariwise to the alveolar space a specialized subset of bronchus-associated macrophages BAM but not the DC are the main subset involved in Ag-uptake in the conducting airways. They re-evaluate the contribution of DC in the initiation of inflammation in an experimental model of allergic airway disease. The authors conduct a very nice set of experiments with different transgenic mice models to selectively deplete DC or macrophages and multi photon imaging of live tissue or 3D reconstruction of thick lung sections.

The authors bring evidence that BAM are potent phagocyte of soluble Ag in specific regions of the lung. Although they show that BAM mediates eosinophil recruitment in a model of allergic reaction, the experiment performed do not support convincingly their contribution to T cell activation.

I raised different points below.

1. One limitation of the study is that the phenotype of BAMs by flow cytometry overlaps with all IM of the lungs, vascular-, nerve associated and pleural macrophages. It is thus difficult to address specifically the function of BAM in the initiation of allergic inflammation as the depletion strategies are not specific of BAM.

The authors focus their imaging study at the branchpoint regions, in collagen rich structure where they observe accumulation of BAM, eosinophils and numerous other CSF1R+ cells.

It is unclear whether the author suggest that these regions represent a more specific route of Ag uptake compared to other regions of the bronchial epithelium. The images of OVA uptake are very convincing but lack quantification to address this point.

The authors could provide a quantification from their imaging study showing the relative proportion of EGFP+ cells with OVA uptake in different regions (branchpoints, airway epithelium and pleura).

The authors claim that soluble Ag is preferentially captured by BAMs but they excluded AM from the flow experiment quantification. As it is important to compare Ag uptake between DC and IM it is also important to see the contribution of AM. The authors should add in figure 4A B the contribution of siglecF+ AM.

2. Eosinophilia is used as the major read out of the macrophage depletion experiments at the beginning of the manuscript and the authors conclude on a non-necessary role of DC to drive allergic recall response and that other subsets may activate T cells. Eosinophil recruitment might be mediated by macrophages but T cell activation mediated by DC. A direct link between effector T cell activation and eosinophilia should be presented at the very beginning to draw such conclusion or maybe postpone this conclusion only after the OT2 experiments. However, the DC compartment is not fully depleted, considering the strong ability of DC to activate T cells compared to macrophages as showed Figure 6, even a small fraction of DC could maintain allergic response.

3. Figure 6C. Proliferation of OT2 cells can be easily reached in the presence of the OVA peptide even with nonprofessional APC. OVA peptide addition is a good control but the stimulation of OT2 cells by freshly purified APC from OVA-sensitized mice and/or with APC loaded with OVA protein is necessary.

4. During live imaging, many cell behaviors can be recorded, the authors need to provide proper quantification of cell interactions in order to identify an overall tendency. What is the proportion of OT2 interacting with EGFP+ cells compared to YFP+? Is the interaction OVA-dependent?

5. Percentages do not prove efficient depletion of cell in DTR system. Lower % could be biased by increased % of another subset. To compare the depletion of the AM in CD11c and CD169 DTR systems the authors should show the absolute number of AM as done for IM Figure 3.

6. Figure 4 There is a significant fraction of CD24+ CD64+ subset. In panel 3C the CD24dimerCD64+ subset is excluded from gating but in Figure 4A it is included in the macrophages. What is this subset exactly?

*Reviewer #2:*

In allergic asthma, parabronchial inflammation occurs within the lung upon inhalation of allergens and activation of Th2 leading to cytokine productions promoting pathogenesis. The study of Tang et al. aimed to identify the cell population responsible for antigen uptake and presentation in the airway and to characterize how this cell population can stimulate a Th2 response locally. The study is based in large parts on usage of two photon microscopy analysis of lungs from a collection of genetically engineered mice to follow myeloid cells at the level of airways in the steady state or after antigen inhalation. The authors found that a population of interstitial macrophages referred to as Bronchus-Associate Macrophages (BAMs) are very efficient for capture and presentation of soluble antigens. These BAMs are located in the steady state at airway bifurcation/branchpoint underneath the epithelium together with Dendritic cells (DCs essentially DC2) and collagen. Interestingly, this corresponds to location where inhaled antigens are likely to be deposited. Accordingly, the authors show that the BAMs exhibit a better ability to uptake inhaled antigens than the DCs. In addition, upon local sensitization, DCs but not BAMs, upregulate CCR7 and migrate to LN while BAMs remain lung resident where they can locally induce allergic inflammation.

The study is well conducted with some elegant experiments and represents a large body of work. The conclusions are of interest. The fact that this population has been missed before due to its intermediate expression of CD11c is very well documented.

I have two main concerns:

1) The authors claim that they have identified a new population of macrophages and try to compare it with the abundant literature on myeloid mouse subsets in their 10 page long discussion. I guess the best way to clearly define a new population for which they have a nice phenotype would be to purify these cells and RNAseq them. This would allow the authors and the readers to actually compare this population to the ones identified in previous studies. RNAseq in bulk is relatively cheap and does not need a lot of cells nowadays.

2) Many nice observations by two photon microscopy are provided without any quantification or performed with too few experimental points to be convincing. "Representative images" that are presented cannot be taken as results since some of the most important conclusions of the study rely on these images. For instance, in Figure 2 there is no quantification of the images of antigen uptake, OVA and HDM (Figure 2D and E). The same holds true for almost all the images presented. I understand that 2 photons microscopy is difficult and time consuming but for some experiments it should be done like the authors did in Figure 1G and H.

*Reviewer #3:*

In this report, Tang and colleagues have used two-photo microscopy to assess in detail the localization and the behavior of CX3CR1-GFP+CD11c+ MHC-II high interstitial macrophages (IM) in vivo. They convincingly identified areas rich in CX3CR1-GFP+ IMs and CD11c-YFP+ DCs in the steady-state lung, particularly in collagen-rich regions near some bifurcating airways. They called the CX3CR1-GFP+ IMs of those areas "Bronchus-Associated Macrophages (BAMs)", and importantly showed that BAMs have a dendritic morphology and can take up soluble antigens more importantly than DCs. BAMs remain lung-resident, are able to process and present soluble antigens, activate T cells in vitro and interact with T cells in an OVA-induced asthma model. These findings are consistent with the idea that BAMs may have previously been overlooked as antigen-presenting cells in the lung. Thus, the title claims that "Bronchus-associated macrophages efficiently capture and present soluble antigens", and such claim is fully supported by the data and well demonstrated. The experiments related to that part are sound, and the report is of great interest and adds to our understanding of lung interstitial macrophage biology by showing that MHC-II high IMs can act as antigen-presenting cells in vivo, as previously suggested in vitro (Chakarov et al., Science, 2019). This part of the work warrants publication in *eLife*.

The concern is related to the second claim of the title, "… for local Th2 cell activation". In my opinion, experimental evidence for that is very weak, and, more globally, the paradigm-shifting idea that BAMs, but not DCs, are necessary for "allergic recall responses in the lung" should, in my opinion, be removed from this report (data related to Figure 3). Indeed, unambiguously demonstrating this would require a huge amount of additional experiments and should be the scope of another entire paper. The main issues related to this part are summarized below:

– No evidence for local Th2 activation has been shown in vivo. Only some data showing that BAMs can trigger T cell proliferation and IL-13 are shown ex vivo, using Th2 polarized effector T cells. A previous report (Chakarov et al., Science, 2019) has shown that CX3CR1high MHChigh IM subpopulations could induce Foxp3+ Tregs ex vivo. Additional intracellular stainings for transcription factors (Foxp3, Gata-3, T-bet, Rorgt) and cytokines (IL-4, IFN-g, IL-17), co-culture experiments with unpolarized naïve, or polarized memory or effector T cells should help deciphering differences in the ability of cDCs vs. BAMs to regulate T cell responses ex vivo. However, additional in vivo proof would also be required.

– The authors employed DT-induced cell depletion in several transgenic animals, in combination with a mouse model of OVA/alum-induced eosinophilia, to support the claim that BAMs, rather than DCs, can serve as antigen-presenting cells to activate local Th2 cell responses in the lung. Weaknesses of this set of experiments are:

1) the paucity of read-outs: only BAL eosinophilia has been measured, while many cardinal features of allergic asthma should have been assessed: peribronchial and perivascular inflammation, mucus production, local Th2 cell recruitment and activation, antigen-specific humoral responses.

2) relevance of the model: most of the recent studies aiming at deciphering the contribution of cDCs in type 2-allergic asthma have used another more relevant model of allergic asthma based on exposure to house dust mite extracts (HDM), a major allergen source in humans. This model should have been used to validate the findings in the OVA/alum "artificial" model. Using a model based on HDM would also allow to decipher the contribution of cDCs vs. BAM in the sensitization phase vs. the recall response.

3) while it seems indeed that BAMs are depleted in CD11c-DTR mice, but not in zDC-DTR mice, while cDCs are depleted in both models, the unambiguous demonstration of the role of cDCs vs. BAM in triggering Th2-cell-mediated allergic responses (either during the sensitization or the challenge phase) would require adoptive transfer of cDCs vs. BAM in DT-treated CD11c-DTR or zDC-DTR animals, to see if that transfer can restore the phenotype affected by DT treatment.

---

## [Author Response]

[Editors’ note: The authors appealed the original decision. What follows is the authors’ response to the first round of review.]

We thank the reviewers and editors for their consideration of our manuscript and constructive feedback. Based on this feedback, we made revisions to the manuscript as outlined below.

We first copy here some key comments from reviewer #3, which we believe captured sentiments shared by the other reviewers:

“Thus, the title claims that "Bronchus-associated macrophages efficiently capture and present soluble antigens", and such claim is fully supported by the data and well demonstrated. The experiments related to that part are sound, and the report is of great interest and adds to our understanding of lung interstitial macrophage biology by showing that MHC-II high IMs can act as antigen-presenting cells in vivo, as previously suggested in vitro (Chakarov et al., Science, 2019). This part of the work warrants publication in eLife.”“The concern is related to the second claim of the title, "… for local Th2 cell activation". In my opinion, experimental evidence for that is very weak, and, more globally, the paradigm-shifting idea that BAMs, but not DCs, are necessary for "allergic recall responses in the lung" should, in my opinion, be removed from this report (data related to Figure 3). Indeed, unambiguously demonstrating this would require a huge amount of additional experiments and should be the scope of another entire paper.”

Based on these comments, we felt that the strength of our manuscript was the characterization of antigen capture and presentation by the BAMs versus DCs and their strategic positioning near the airways. However, clearly a weakness of the manuscript was that the data did not convincingly demonstrate that BAMs rather than DCs promoted local Th2 cell activation in the lung. We agree with this assessment. Due to limitations of depletion strategies, which have relatively poor specificity and limited efficacy, it would be very difficult to definitively establish that DCs are not involved in antigen presentation to Th2 cells in the lung. Indeed, we had never intended to exclude this possibility, but rather to show that BAMs are efficient at capturing and presenting antigen, are capable of activating Th2 cells, and remain lung resident. We note that we had included in our manuscript (lines 418-420, formerly 430-432) the following text: “These observations suggest that both BAMs and DCs can serve as APCs for the local activation of effector T cells in lung peribronchial regions.” In the revised manuscript, we have modified the latter part of the title to state that BAMs “are capable of local Th2 cell activation.” Specifically testing the functional role of BAMs has been difficult due to the limited ability to target this population versus other macrophage populations and DCs as well as the replenishment of depleted BAMs by monocytes. We agree with Reviewer #3 that such investigations would require a large amount of additional experiments and would be the scope of a different paper, and at present such a study may not even be feasible until better tools are developed.

Accordingly, given the concerns voiced by both Reviewers 1 and 3 regarding the functional data in an asthma model, we have removed Figure 3 (with the different DTR mice) and the associated supplementary figures from the manuscript, and modified the text accordingly. We have also refocused the manuscript on further characterization of the BAMs and comparison to DCs. Some highlights of what we have added to the manuscript include:

Another concern raised by the reviewers was that insufficient quantification was provided for imaging data. We appreciate this concern and the value of quantitative data. It also must be recognized that for such quantifications to be done in a robust manner with large 3D image stacks collected by two-photon microscopy requires an extensive, concerted effort with careful image analysis to avoid artifacts. We have therefore focused on adding quantification specifically regarding key findings in the manuscript. Specifically, we have added:

Figure 2 – We had added new data on fluorescent OVA uptake per individual cell (comparing BAMs to DCs) and then the geometric mean fluorescence intensity per cell type in each imaging field (Figure 2E). We have also added data on individual cell shapes (sphericity and length of longest axis) in Figure 2G/H in addition to the existing median/mean data per imaging field.

Figure 7 – We have added an analysis of the interaction times of individual OT-II T cells with BAMs in Figure 7G.

We also recognize that two-photon microscopy, while a very powerful technique, still gives a limited window into specific regions of the lung that are being imaged. Significant variability exists in the distribution of inhaled antigen throughout the lung airways, as well as the appearance of different lung regions. We nonetheless feel confident by imaging these regions and cell types for over a decade that the image examples we have provided here are representative of what can be observed using this approach. We also note that our extensive flow cytometric analyses quantify many of the characteristics of BAMs in terms of antigen uptake and processing as well as marker expression, which complements our findings by microscopy.

We believe that the above revisions address most of the comments of the reviewers. Some additional specific responses to reviewer comments are noted below:

Reviewer #1: As it is important to compare Ag uptake between DC and IM it is also important to see the contribution of AM.

We agree that AMs take up substantial amounts (in fact most) of the antigen that is inhaled. However, AMs are (1) located at a different site (alveoli) rather than near the bronchial airways where inflammation occurs in asthma, and (2) have been shown in previous studies to poorly stimulate CD4 T cells. We confirmed that peptide-MHC-II complexes were undetectable on AMs (Figure 7A). We thus think that while AMs may have a very important phagocytic role in the lung, effectively ‘cleaning’ up the airspaces in the alveoli, that the role we have identified here for BAMs in antigen capture and presentation is important in the context of T cell activation. Accordingly, our analyses have been focused on BAMs and DCs which are located proximal to the airways and process and present antigen peptides on MHC-II.

Reviewer #1: Figure 6C. Proliferation of OT2 cells can be easily reached in the presence of the OVA peptide even with nonprofessional APC. OVA peptide addition is a good control but the stimulation of OT2 cells by freshly purified APC from OVA-sensitized mice and/or with APC loaded with OVA protein is necessary.

We chose to use OVA peptide for these sorting experiments because it was difficult to get adequate T cell activation for our analysis with OVA protein taken up in vivo after enzymatic lung digestion and cell sorting of APCs (this was true of any APC subset including DCs). Given the number of steps (and hours) required to isolate these cells to high purity, we felt this was a technical limitation of the assay. This is why we separately confirmed that BAMs present peptide-MHC-II complexes on their cell surface derived from inhaled antigens and express costimulatory molecule (Figures 7A,B). We still think these cell sorting data are valuable to show that BAMs are capable of activating Th2 cells in vitro.

Reviewer #1: During live imaging, many cell behaviors can be recorded, the authors need to provide proper quantification of cell interactions in order to identify an overall tendency. What is the proportion of OT2 interacting with EGFP+ cells compared to YFP+? Is the interaction OVA-dependent?

We appreciate the reviewer’s suggestion. We have focused on the quantification of interactions of OT-II cells with BAMs given the revised nature of the manuscript, and showed that the interactions are indeed dependent on inhaled OVA as shown in Figure 7G.

Reviewer #1: Figure 4 There is a significant fraction of CD24+ CD64+ subset. In panel 3C the CD24dimerCD64+ subset is excluded from gating but in Figure 4A it is included in the macrophages. What is this subset exactly?

We agree with the reviewer that there is variation in CD24 expression among CD64+ cells, but we also note that CD24 expression is lower in these cells compared with DCs which have high CD24 expression. The inclusion of these markers has been helpful to gate cDCs versus IMs in past studies and accordingly were included here. We also note that CD64 expression correlated with MerTK, as expected from past studies of macrophages. The nature of this variation in CD24 expression in CD64+ cells is unclear, but is beyond the scope of our manuscript. The former Figure 3C was removed from the manuscript. Our microscopy data confirm that CD64 is a relevant marker distinguishing BAMs from DCs.

Reviewer #1: The statement "Microscopy detects a comparably narrower dynamic range of fluorescent signals than flow cytometry, and thus we most likely only visualized CX3CR1-GFPbright IMs that carried the most OVA" is somehow incorrect. Each pixel of an image is indeed scaled on a narrower band of resolution but the image quantification should be considered at the scale of the object. Figure 2A and E shows that free or very small vesicules containing OVA-FL and HDM-FL can be detected suggesting that even YFP+ cells with small amount of antigen should be seen on the images. This point should be considered for image quantification.

We thank the reviewer for bringing up the important issue of quantifying the uptake of fluorescent antigen. First to clarify our comments here, there is a difference between the size of a region containing a fluorescent signal, versus the absolute intensity of the fluorescence signal. The absolute intensity of the signal by two-photon microscopy is affected by several factors including laser power, detector gain, and tissue scattering of light. These properties are all quite different than the analysis of single cells by flow cytometry. In our experience, there is a much more limited range of what can be visualized by two-photon microscopy than by flow cytometry and again this depends on the settings of the microscope. There is also a limitation to what can be visualized when looking at an image by eye, which depends on the display range of the image. To indeed illustrate how different linear adjustments can lead to different conclusions of whether signal is present or absent, we show in Figure 2—figure supplement 2 two different adjustments to the same image. Our statement above was meant to indicate how based on our typical settings, we most likely only visualized CX3CR1-GFP^bright^ IMs that carried the most OVA. Related to the reviewer’s comment, we recognize that image quantification is not the same as visualization by eye. We have therefore added a careful quantification of OVA uptake in BAMs versus DCs (quantified as the sum of fluorescence intensity per cell) from our microscope images in Figure 2E in the revised manuscript. These data confirm that weak OVA fluorescence signal was detected by microscopy in CD11c-YFP+ DCs, and are in line with our observations by flow cytometry.

Reviewer #1: "Alveolar macrophages, with low MHC-II expression and a high degradative capacity, showed negligible Y-Ae staining (Figure 6—figure supplement 1), confirming these are not the relevant APCs for CD4 T cells" Indeed it is unlikely that AM are good APC but the absence of detectable Y-Ae staining cannot preclude antigen presentation capacity. Ag presentation assay of AM sorted from OVA-sensitized mice with OT2 for instance could confirm.

Respectfully, to our knowledge, an essential feature of APCs for CD4 T cells is the presentation of peptide complexes on MHC-II. Since (1) AMs have low MHC-II expression, (2) are known to be highly degradative, and (3) had undetectable peptide-MHC-II complexes on their cell surface with the well-characterized Y-Ae antibody, whereas we could detect these peptide-MHC-II complexes on BAMs and DCs, we believe our conclusion is reasonable. Please see above for other notes on why we have not focused on AMs in this study. We also note that other studies have indicated that AMs may have a suppressive effect on the immune response.

We agree with this comment and provided new images which give more information about MHC-II and CD64 expression around the airway. This is now Figure 3C.

In summary, we believe the revisions to the manuscript address the requests in the decision letter from the editors: